# Different honesty conceptions align across US politicians' tweets and public replies

Fabio Carrella [1] ✉, Segun T. Aroyehun[2], Jana Lasser [3,4], Almog Simchon [5], David Garcia[2,4] & Stephan Lewandowsky [1,6]

Recent evidence shows that US politicians' conception of honesty has undergone a bifurcation, with authentic but evidence-free "belief-speaking" becoming more prominent and differentiated from evidence-based "fact-speaking". Here we examine the downstream consequences of those two ways of conceiving honesty by investigating user engagement with fact-speaking and belief-speaking texts by members of the US Congress on Twitter (now X). We measure the conceptions of honesty of a sample of tweets and replies using computational text processing, and check whether the conceptions of honesty in the tweets align with those in their replies. We find that the conceptions of honesty used in replies align with those of the tweets, suggesting a "contagion". Notably, this contagion replicates under controlled experimental conditions. Our study highlights the crucial role of political leaders in setting the tone of the conversation on social media.

Online misinformation is becoming an increasing concern for democracies at a time when social media is widely used for political communication and news consumption. Misinformation can take many forms, from conspiracy theories and propaganda to false news and deep fakes[1]. Misinformation can have adverse effects on a variety of social issues, such as undermining trust in scientific and academic institutions, as in the cases of global warming and vaccinations[2,3], or stoking political polarization and cynicism[4,5].

Misinformation can spread unintentionally when communicators believe incorrect information, or intentionally as disinformation to promote certain viewpoints or agendas. An example is Donald Trump's false claims of election irregularities, which encouraged the January 6, 2021 Capitol riots[6,7]. Among his followers, the belief that the 2020 election was stolen appeared to be a genuine belief resistant to interventions that typically separate belief from casual acquiescence[8]. Notably, Trump's presidency was characterized by a lack of veracity (the Washington Post classified 30,573 of his claims as false or misleading during his presidency). Nevertheless, many of his followers not only supported him during his presidency but also considered him to be honest[9]. This disconnect between accuracy and politicians' endorsement by voters has also been shown in experiments involving the

American public in which several of Trump's false claims were corrected[10,11]. In these studies, participants' feelings and voting intentions were unaffected by corrections even though the correction reduced the strength of beliefs in specific falsehoods.

One possible explanation for this divergence between the popularity and accuracy of a politician involves the finding that, when social groups perceive that they lack representation in the political system or are otherwise discarded by a political establishment, an overtly lying demagogue can appear to be an authentic champion of the people who speaks the suppressed truth[12]. This suggests that the act of speaking one's mind by a politician—a skill at which Trump excelled[13]—is considered a better indicator of honesty by segments of the population than factual truthfulness, and that even false statements can be considered honest if they stem from authentic and sincerely expressed beliefs.

As a consequence, honesty ceases to be a monadic concept used to judge people based on evidence (i.e., a person is either lying or telling the truth based on facts) and instead becomes a dyadic construct, where adherence to truthfulness and commitment to personal beliefs, respectively, coexist as distinct aspects of honesty. This dyadic conceptual model of honesty[9] (see also ref. 14) involves two

[1]School of Psychological Science, University of Bristol, Bristol, UK. [2]Department of Politics and Public Administration, University of Konstanz, Konstanz, Germany. [3]IDea_Lab, University of Graz, Graz, Austria. [4]Complexity Science Hub, Vienna, Austria. [5]Department of Psychology, Ben-Gurion University of the Negev, Beer Sheva, Israel. [6]Department of Psychology, University of Potsdam, Potsdam, Germany. ✉e-mail: fabio.carrella@bristol.ac.uk

components, known as "fact-speaking" and "belief-speaking"[15]. The first emphasizes the accuracy of a statement and aims to convey the true state of affairs. The second prioritizes genuine and authentic expression of beliefs, focusing more on a person's emotional or mental state rather than the objective state of the world[16]. Belief-speaking and fact-speaking refer to rhetorical frames that reflect a person's underlying conception of what they consider to be honest[17–19]. It follows that neither belief-speaking nor fact-speaking are ineluctably tied to the truthfulness of the information being communicated. A person can use belief-speaking while still conveying accurate statements, just as they can employ fact-speaking to camouflage their falsehoods. This fluidity arises from the fact that these two aspects of honesty can be viewed as adaptable constructs of discourse that can be readily adjusted and transmitted to others depending on circumstances. Politicians may strategically opt for one particular frame over the other to encourage viewers to tune into their own characterization of reality and obtain a desired outcome[20–22]. If the audience wishes to counter these narratives, they must invest effort[23,24].

A recent analysis of the public speech on Twitter (now X) by all members of the U.S. Congress between 2011 and 2022 identified the presence of those two conceptions of honesty, belief-speaking and fact-speaking[15]. The analysis examined 4 million tweets and identified the prevailing conception of honesty reflected in each tweet. Illustrative tweets are shown in Fig. 1.

For both parties, both belief-speaking and fact-speaking increased considerably after Trump's election in 2016. When the content of tweets was related to the quality of news sources they linked to, a striking asymmetry between the two parties and the honesty components emerged. For members of both parties, the more a tweet expressed fact-speaking, the more likely it was to link to a trustworthy source, as ascertained by NewsGuard ratings. By contrast, for belief-speaking, there was a striking association between increased belief-speaking and lower trustworthiness of sources for Republicans (a smaller association was observed for Democrats)[15]. The findings are compatible with the idea that a distinct conception of honesty that emphasises sincerity over accuracy can be used by politicians as a gateway to the sharing of low-quality information, seemingly without paying an electoral or political price. Additionally, appeals to an intuition-based epistemology by populist leaders can further solidify the social identity of their supporters, transforming the sharing of

misinformation into a marker of group membership and a preference for gut instincts over science-based claims, which in turn sets the stage for the proliferation of further falsehoods[25].

Here we examine the downstream consequences of political communications using these two distinct conceptions of honesty by studying the conversations between politicians and the public on Twitter. Although the platform is now known as X, we use Twitter throughout this study as the data were collected prior to the name change. Platforms such as Twitter have opened up novel avenues for political agenda-setting, exerting a discernible impact on society, both in positive and negative ways. Positive instances of expression of political leadership online can be observed in consistent communication during crises and the promotion of increased transparency and accountability[26,27]. For example, in New Zealand during the COVID-19 pandemic, then Prime Minister Jacinda Ardern's communicative approach, defined as empathetic yet transparent, played a pivotal role in mitigating the virus's threat[28].

By contrast, negative leadership traits manifest through actions such as spreading false information, employing offensive language, and manipulating the public's political agenda[29,30]. A pertinent illustration arises from the U.S., where studies suggest that Donald Trump might have strategically used Twitter to divert media attention away from topics he perceived as personal threats[31]. Although the intentionality behind this diversion cannot be definitively established, other studies have found significant linguistic differences between Trump's factually correct and incorrect tweets[32] (see also ref. [33]), implying that these tweets are unlikely to be random errors and might have been crafted more systematically.

Trump's presidency has also been associated with an increase in affective polarization between the parties[34]. Affective polarization arises when political differences become deeply entrenched and emotional responses dominate attitudes towards in- and out-group members[35]. In recent years, Republicans have seen a significant shift in their party's image, with an increase in words like "patriotic," "loyal," and "Americans." While causality remains to be established, this shift may be influenced by Republican leaders and their nationalistic rhetoric, reflecting a broader trend towards "us-versus-them" thinking in partisan politics[34], which in turn exacerbates affective polarization[4,36,37]. Additionally, increasing affective polarization has been causally linked to belief in misinformation that is favouring the

**Fig. 1 | Examples of tweets from Democrat (blue) and Republican (red) politicians characterized by high belief-speaking (top) or fact-speaking (bottom) scores.** The scores were elaborated in ref. 15 and consist of the cosine similarities between the word embeddings of each tweet and two validated dictionaries. See "Methods" for more details.

political in-group[38]. Conversely, reducing affective polarization also reduces the strength of belief in partisan-aligned misinformation[38].

How, then, do users engage with fact-speaking and belief-speaking texts from US politicians on Twitter? To answer this question, we compiled a corpus of conversations (i.e., tweets and their replies) between Twitter users and US politicians, focusing on two aspects. First, the association between politicians' and repliers' narratives: when a conversation is "seeded" by belief-speaking or fact-speaking, do users' responses align with the chosen view of honesty? Second, we checked whether the presence of one or the other honesty component in a seed (i.e., the politician's original tweet) was associated with the affectively polarized language of its replies.

To foreshadow briefly, our results show that (i) there is an honesty "contagion" in place because the presence of a belief-speaking [fact-speaking] component in a seed is associated with a belief-speaking [fact-speaking] component in its replies, and (ii) fact-speaking seeds seem to be negatively correlated with effectively polarized language in the replies, whereas belief-speaking seeds show the opposite trend. Notably, both results are robust to several potential confounding variables such as tweet topics and authors, as well as partisanship. Because our analysis of naturalistic speech was necessarily correlational, we additionally conducted an experiment in which we exercised control over the content of the seeds. In the experiment, participants were asked to write free-form replies to our seeds, and we examined whether the tenor of their replies aligned with the honesty conception in the seeds. We replicated the "contagion", suggesting that the seeds' effects are causal, although in the experiment, the evidence for fact-speaking seeds reducing affective polarization was more tentative.

## Results

### Sample of conversations and honesty components identification

We created our main dataset by randomly selecting 20,000 tweets from a bigger corpus of tweets employed in ref. 15. The corpus included tweets from US Congress members posted between January 2016 and March 2022. The sampled tweets served as starting points (i.e., "seeds") for the collection of public interactions. We were only able to collect conversations from 13,169 seeds due to a variety of circumstances (see "Methods" for details). The total number of replies across the conversations was 331,373. Following multiple filtering procedures, we were left with 97,510 responses to 10,164 seeds from 728 US politicians (Democrats = 386, Republicans = 342). See "Methods" for more details on the curation of replies.

Once the dataset was created, we needed to identify the type of honesty construct—belief-speaking or fact-speaking—that was prevalent in each of the texts (i.e., seeds and replies) collected. The methodology used for the identification of the two honesty constructs has been fully described in ref. 15 and is briefly summarized here.

To identify belief- and fact-speaking in the texts, we first compiled two sets of keywords that we believed represented each category. These keyword sets were computationally expanded and validated through a series of surveys, as described in ref. 15. Subsequently, we employed word embeddings (using GloVe[39]) to derive contextual representations of each keyword. We then averaged these representations to obtain two distinct embeddings, one for belief-speaking and the other for fact-speaking, so that each represented a distributed dictionary[40].

We used word embeddings in an analogous manner to extract contextual representations for each tweet in our dataset. We next computed the proximity (i.e., the cosine similarity) between the average embedded representations of each tweet and those of our dictionaries. This process produced two similarity scores for each tweet, which conveyed the degree of belief-speaking and fact-speaking in the tweets.

To consolidate these two scores, we independently standardized them and then subtracted belief-speaking from fact-speaking

similarity. This resulted in a FmB (Fact-minus Belief) score, which identifies texts as leaning towards belief-speaking when FmB < 0 and texts expressing fact-speaking when FmB > 0. See Fig. 1 for examples of texts with low FmB score (top row) and high FmB score (bottom row).

### Conversational alignment of honesty constructs

Our primary analysis focused on the alignment of honesty components between Twitter seeds and their replies (see "Methods" for further details). We performed a linear mixed-effects model regression, where the honesty score of the replies ($FmB_r$) was the dependent variable, and the honesty score of the seeds ($FmB_s$) was the main independent variable. The latter was fully crossed with two further predictors, namely the party of the politician who wrote the seed (Party), and the estimated ideology of the person who replied to the seed ($I_{score}$), which was inferred from the public figures followed on Twitter by each repliers (for details on how the ideology was extracted, see Methods). Additionally, we used a measure of affectively polarized language in the seeds ($Pol_s$, see "Methods" for details) as a proxy for aspects such as toxicity and incivility, aiming to disentangle these from belief-speaking expressions. A further analysis accounting for instances of positive and negative emotions in the seeds is reported in the Supplementary Information (Section S1). We also included two random effects, namely the seeds nested within their authors, and the topic of the seeds (see "Methods" for details on topic modeling). Repliers were not included as random effects as we only kept one reply per respondent. Random effects of repliers were, therefore, automatically modeled by the random effect of seeds nested within authors. For further details on the regression and its variables, see "Methods".

The results of this component alignment analysis suggested a positive association between components across seeds and replies for both parties. $FmB_s$ was positively associated with $FmB_r$ ($t(97, 510) = 19.863$, $p < 0.001$, $\beta = 0.075$, 95%CI = [0.067, 0.082]). This association suggests the potential existence of a "contagion", where the original seed influences the tone of the replies: if a politician tweeted belief-speaking information (FmB < 0), the replies were also accentuated in the direction of belief-speaking. Likewise, a fact-speaking seed (FmB > 0) elicited additional fact-speaking in the replies.

This relationship is illustrated in Fig. 2, which shows the number of replies seeds received, separated by FmB score quartiles. A low quartile (i.e., Q1) indicates belief-speaking, while a high quartile (i.e., Q4) denotes fact-speaking. For example, the left-most column in Panel A shows that Democrat seeds with high belief-speaking received 3340 belief-speaking replies and 1889 fact-speaking replies. Similarly, the right-most column in Panel B shows that Republican fact-speaking seeds received 4740 fact-speaking replies and 2913 belief-speaking replies. Both panels demonstrate that belief-speaking seeds attract more belief-speaking replies and fewer fact-speaking replies, and vice versa.

Further predictors, such as the estimated ideology of the replier and the party of the politicians who wrote the seeds showed a significant negative correlation with the honesty scores of the replies ($t(97, 510) = -8.006$, $p < 0.001$, $\beta = -0.017$, 95%CI = [-0.021, -0.013]; $t(97, 510) = -3.435$, $p < 0.001$, $\beta = -0.025$, 95%CI = [-0.040, -0.011]). This suggests that, regardless of the honesty framing present in the seed, replies in our dataset have a higher chance of tending towards a belief-speaking framing when the replier is more conservative or when the original seed is written by a Republican politician. The presence of effectively polarized language in the seeds ($Pol_s$) is also significant and negatively correlated with $FmB_r$ ($t(97, 510) = -5.690$, $p < 0.001$, $\beta = -0.015$, 95%CI = [-0.021, -0.010]), suggesting that a higher frequency of such language in a seed is associated with a greater occurrence of belief-speaking in replies.

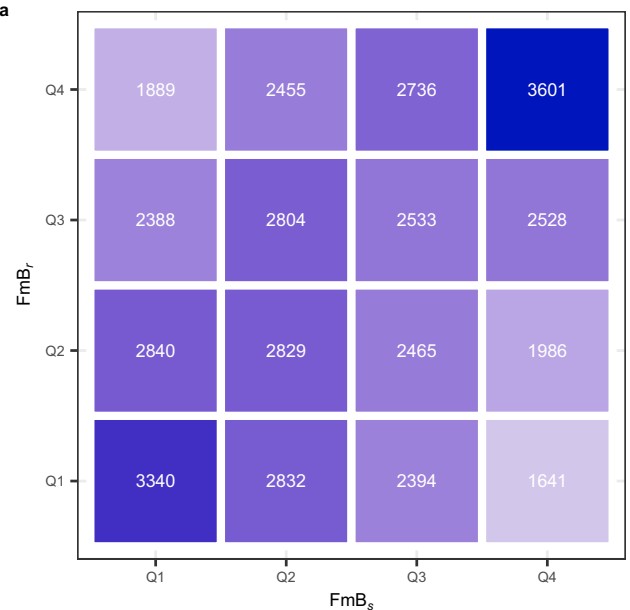

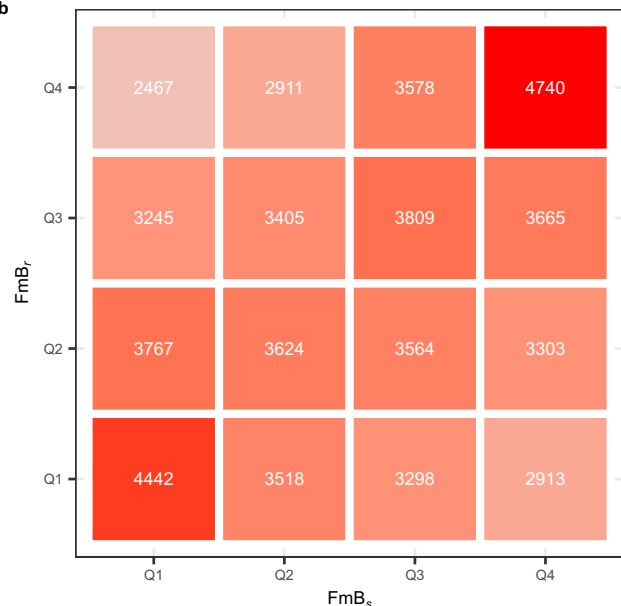

**Fig. 2 | Relationship between honesty components in seeds and replies: each tile represents the number of replies received by seeds in each FmB score quartile.** In both panels, the x-axes represent the quartile of the seeds $FmB_s$, and the y-axes represent the quartile of the replies $FmB_r$. For both the x- and y-axes, Q1 indicates texts with a low FmB score, thus characterized by belief-speaking, whereas Q4 indicates texts with a high FmB score, which signals fact-speaking. Panel **a** shows frequencies for seeds written by Democrat politicians, whereas Panel **b** shows frequencies for seeds written by Republican politicians. The gradient tonality of the color represents the number of replies to seeds in each FmB quartile, with darker tones indicating more replies and lighter tones indicating fewer replies.

We also found the honesty scores of the replies to be significantly predicted by the two- and three-way interactions between the honesty scores of the seeds, the ideology of the replies, and the party affiliation of the seed's author. The negative coefficient for the interaction between the honesty scores of the seeds and the politicians' affiliations suggests that the influence of the seed component on the reply component is stronger for seeds written by Democrats ($t(97, 510) = -3.840$, $p < 0.001$, $\beta = -0.019$, 95%CI = [−0.029, −0.009]). By contrast, the interaction between the ideology of the replier and the party of the politician who authored the seed presents a positive coefficient, implying a cross-partisan relationship between the two variables ($t(97, 510) = 12.379$, $p < 0.001$, $\beta = 0.035$, 95%CI = [0.029, 0.040]). This means that users tend to reply with more belief-speaking when addressing seeds written by politicians of the opposing party.

Finally, we also found a small but significant effect of Date on the honesty scores of the replies ($t(97, 510) = -4.004$, $p < 0.001$, $\beta = -0.011$, 95%CI = [−0.017, −0.006]). This suggests that replies in our sample tended to present more belief-speaking than fact-speaking over time. The position (Pos) of the reply in the conversation in chronological order is also significant, but its impact is essentially negligible ($t(97, 510) = -3.462$, $p < 0.001$, $\beta = -0.006$, 95%CI = [−0.010, −0.003]). Further details for the model and its variables are presented in Table 1 and in "Methods".

Figure 3 illustrates the overarching three-way interactions between the honesty scores of the seeds, the ideology scores of the repliers, and the political parties of the seed authors ($t(97, 510) = 2.277$, $p = 0.022$, $\beta = 0.006$, 95%CI = [0.001, 0.012]). It is evident how the honesty scores of the replies are positively related to those of the seeds. The slopes in the left panel are generally less flat than those in the right panel, supporting what we observed in the two-way interaction between $FmB_s$ and the Party, namely that the contagion is stronger when seeds are written by Democrat politicians. The graph also suggests that the contagion is higher when there is an affiliation between the politicians' parties and the repliers' ideology. More precisely, more liberal repliers (blue line in the left panel) appear more prone to contagion when addressing Democrats' seeds, and the same holds true for

more conservative users (red line in the right panel) who reply to Republicans.

Based on these findings, we aimed to further investigate the interplay between the contagion and the political similarity between repliers and politicians. We examined whether the positive relationship between the honesty components of seeds and replies persisted among users who responded to both belief-speaking and fact-speaking messages from the same politician(s). As our main analysis included only one reply per replier, one possible explanation of our results could be that users aligned themselves according to their preferred honesty components. In other words, individuals who preferred belief-speaking or fact-speaking frames, respectively, might interact primarily with politicians employing those specific frames.

To address this, we refined our dataset of replies by selecting replies from users who engaged with both belief-speaking and fact-speaking seeds from the same politician(s). This resulted in a dataset consisting of $N = 2105$ repliers and $N = 2350$ unique pairs of repliers and politicians (the same replier could respond to multiple politicians). Considering that belief-speaking and fact-speaking were gauged along a continuous spectrum, we classified seeds into either category based on whether they fell within the first or fourth quartile of the FmB score. Subsequently, we calculated the average FmB scores for their replies, resulting in two scores for each replier: one indicating the FmB score of their replies to belief-speaking seeds and another indicating the FmB score of their replies to fact-speaking seeds. Finally, we conducted two-sided paired t-tests by entering the FmB scores of each replier for the two different types of seeds into the analysis, examining whether there was a significant difference between the scores. The t-tests revealed a significant difference between the two sets of FmB scores ($t(2349) = -13.132$, $p < 0.001$, $d = -0.271$, 95%CI of difference in means = [−0.152, −0.112]). Repliers exhibited higher FmB scores in response to fact-speaking seeds ($M = 0.067$, SD = 0.400) compared to belief-speaking seeds ($M = -0.065$, SD = 0.362) of the same politician. Because all parties involved in the conversations were the same and only the nature of the seed varied, this analysis rules out the possibility that our primary result merely reflected a self-selection effect.

**Table 1 | Results of the generalized linear mixed-effects models described in Equation (2) and (3), with replies' FmB scores (FmB_r, left) and affective polarization scores (Pol_r, right) as dependent variables**

| | Dependent variable: | |
|---|---|---|
| | FmB_r (1) | Pol_r (2) |
| FmB_s | 0.075 | −0.009 |
| | (0.067, 0.082) | (−0.011, −0.006) |
| | $p < 0.001$ | $p < 0.001$ |
| I_score | −0.017 | 0.013 |
| | (−0.021, −0.013) | (0.012, 0.014) |
| | $p < 0.001$ | $p < 0.001$ |
| PartyRepublican | −0.025 | 0.020 |
| | (−0.040, −0.011) | (0.016, 0.024) |
| | $p = 0.001$ | $p < 0.001$ |
| Pol_s | −0.015 | 0.012 |
| | (−0.021, −0.010) | (0.011, 0.014) |
| | $p < 0.001$ | $p < 0.001$ |
| Date | −0.011 | 0.002 |
| | (−0.017, −0.006) | (0.001, 0.004) |
| | $p < 0.001$ | $p = 0.006$ |
| Pos | −0.006 | 0.000 |
| | (−0.010, −0.003) | (−0.001, 0.001) |
| | $p = 0.001$ | $p = 0.567$ |
| FmB_s:I_score | −0.003 | −0.000 |
| | (−0.007, 0.001) | (−0.002, 0.001) |
| | $p = 0.202$ | $p = 0.471$ |
| FmB_s:PartyRepublican | −0.019 | 0.005 |
| | (−0.029, −0.009) | (0.003, 0.008) |
| | $p = 0.001$ | $p = 0.001$ |
| I_score:PartyRepublican | 0.035 | −0.023 |
| | (0.029, 0.040) | (−0.024, −0.021) |
| | $p < 0.001$ | $p < 0.001$ |
| FmB_s:I_score:PartyRepublican | 0.006 | 0.001 |
| | (0.001, 0.012) | (−0.000, 0.003) |
| | $p = 0.023$ | $p = 0.118$ |
| Constant | 0.042 | 0.002 |
| | (0.026, 0.058) | (−0.004, 0.007) |
| | $p < 0.001$ | $p = 0.553$ |
| Observations | 97,510 | 97,510 |

Estimates are reported together with their 95% confidence intervals. All predictors were scaled by subtracting their means from their values and dividing the result by their standard deviations. The regression was performed with the function lmer from the R library lme4[67].

Figure 4 provides a qualitative perspective of the honesty conceptions by showing how words from our dataset of replies are arranged on a two-dimensional plot. The x-axis represents the ideology spectrum of repliers, and the y-axis represents the FmB_r scores. Each dot is a unigram from the replies corpus, with coordinates ranging from −1 to 1, indicating the representativeness (or *keyness*) of that term along the ideology dimension (less vs. more conservative) and the honesty dimension (fact-speaking vs. belief-speaking). A word with coordinates of 0 is equally frequent across both dimensions. By contrast, a dot in the top-right corner indicates a term frequent in fact-speaking replies by conservative repliers, while a dot in the bottom-left indicates a term frequent in belief-speaking replies by liberal repliers.

The scatterplot highlights how belief-speaking is the honesty component that most convey inter-partisan communication between

users. In the bottom-right corner, we find keywords characterizing belief-speaking replies from more conservative users, many of which refer to the opposite party (e.g., "democrats", "biden", "obama", "dem", "liberal"). Similarly, the bottom-left corner shows the same behavior from more liberal users (e.g., "trump", "republican"). This aspect on its own would only represent further evidence of opinion polarization in the political debate on social media. However, what is problematic is the presence of other keywords that denote affective polarization, such as "hate", "damn", "dumb", "traitor", and so on. It is worth noting how the majority of such keywords tend towards the center of the ideological axis, indicating they are used by both sides of the spectrum to an almost equal extent. By contrast, fact-speaking keywords, depicted on the top part of the graph, mostly refer to social issues. At a glance, more conservative users seem concerned about regulation (e.g., "border", "fraud", "control", "law", "illegal"), and COVID-19 ("vaccine", "virus"), whereas more liberal users mostly refer to aid measures (e.g., "wage", "help", "health", "healthcare").

### The relation between affective polarization and honesty

We next considered how effectively polarized language in the replies relates to the presence of the two honesty components in the seeds. To do so, we computed polarization scores for both replies (Pol_r) and seeds (Pol_s) from Twitter using the same approach we employed to identify the honesty components. That is, we extracted an averaged word embedding representation from an affective polarization dictionary[41], and calculated its cosine similarity with the averaged word embedding representation of each of the texts. This resulted in a polarization score for each text (see "Methods" for further details).

Then, we performed a linear mixed-effects regression with the affectively polarized language of the replies (Pol_r) as the dependent variable, and the honesty score of the seeds (FmB_s) as the main independent variable. As with the alignment analysis, FmB_s were entered in a fully crossed three-way interaction with the party of the seed author, and the ideology of the replier. We also included the affectively polarized language of the seeds as an independent variable to control for its effect. Finally, we included the same two random effects, the seeds nested within their authors and the topic of the seeds. For further details on the regression and its variables, see "Methods".

The results for this analysis highlight a negative relationship between the affectively polarized language in the replies and the honesty component in the seeds ($t(97, 510) = −7.618$, $p < 0.001$, $\beta = −0.009$, 95%CI = [ −0.011, −0.006]). This indicates that polarizing language is less frequent when seeds present a fact-speaking frame. Moreover, the Supplementary Information (Section S2) shows how this association is even stronger in replies to controversial topics. However, it's worth noting that while statistically significant, the estimated effect size is relatively small compared to other predictors.

Affectively polarized language in the seeds, by contrast, was associated with a positive coefficient ($t(97, 510) = 15.224$, $p < 0.001$, $\beta = 0.012$, 95%CI = [0.011, 0.014]), suggesting that a seed containing polarizing terms will attract more polarizing discourse in the reply it receives. When considering partisanship, both the ideology of the repliers and the party of the seed authors have positive significant correlations with the affectively polarized language in the replies ($t(97, 510) = 18.977$, $p < 0.001$, $\beta = 0.013$, 95%CI = [0.012, 0.014]; $t(97, 510) = 10.414$, $p < 0.001$, $\beta = 0.020$, 95%CI = [0.016, 0.024]), with the latter having the higher coefficient (even higher than Pol_s). This suggests that replies are more likely to contain polarizing language especially when addressed in response to a Republican politician's seed, as well as when written by a more conservative replier.

Significant interactions were observed between FmB_s and Party ($t(97, 510) = 3.689$, $p < 0.001$, $\beta = 0.005$, 95%CI = [0.003, 0.008]), and between I_score and Party ($t(97, 510) = −24.807$, $p < 0.001$, $\beta = − 0.023$, 95%CI = [−0.024, −0.021]). The former, illustrated in Fig. 5a, shows that polarizing language in response to Democrat seeds decreases more

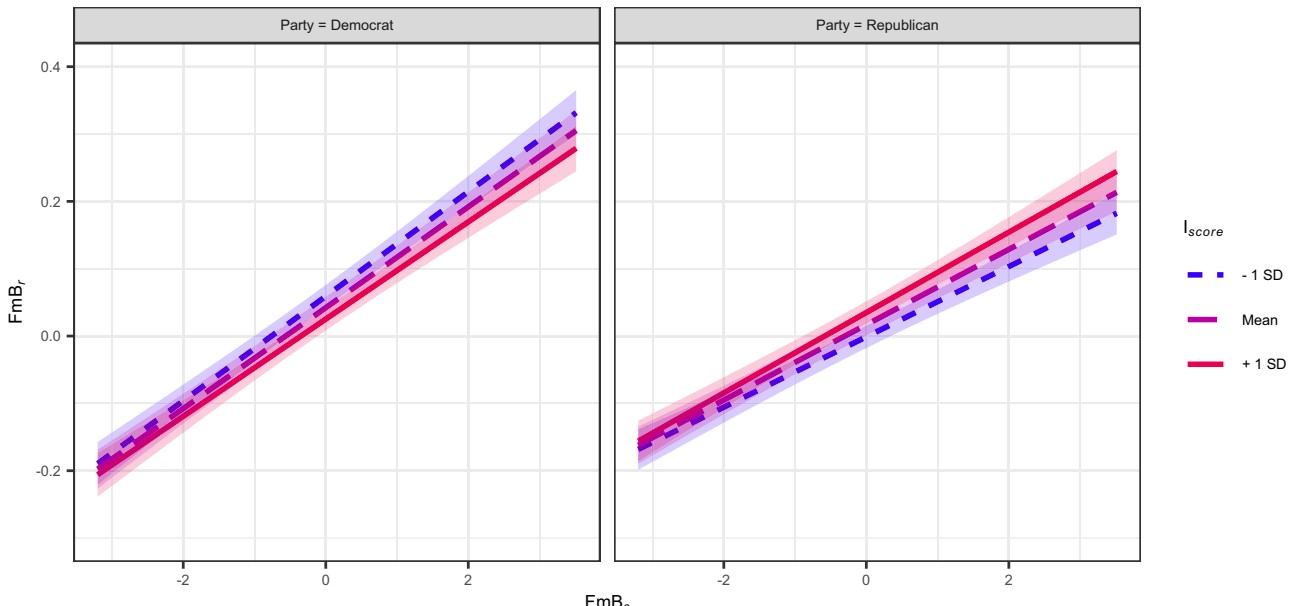

**Fig. 3 | Effects of the three-way interactions between the honesty scores of the seeds (FmB$_s$), the ideology of the repliers ($I_{score}$), and the party of the politicians who wrote the seeds (Party) on the honesty scores of the replies (FmB$_r$).** The estimates are derived from the regressions described in Equation (2) (see "Methods"). The figure is divided into two panels: the left-hand panel considers seeds written by Democrat politicians, whereas the right-hand panel focuses on seeds written by Republican politicians. The $x$-axes in both panels represent the FmB scores of the seeds. The $y$-axes in both panels show the FmB scores of the replies. In both cases, a value of FmB < 0 signals belief-speaking texts, whereas a value of FmB > 0 indicates fact-speaking texts. The $I_{score}$ scoreline types and colors are assigned based on their relationship to the mean and standard deviation of $I_{score}$. A value of −1 standard deviation from the mean is represented in blue, indicating repliers who tend toward more liberal views, whereas a value +1 standard deviation from the mean is depicted in red, indicating repliers characterized by a more conservative partisanship.

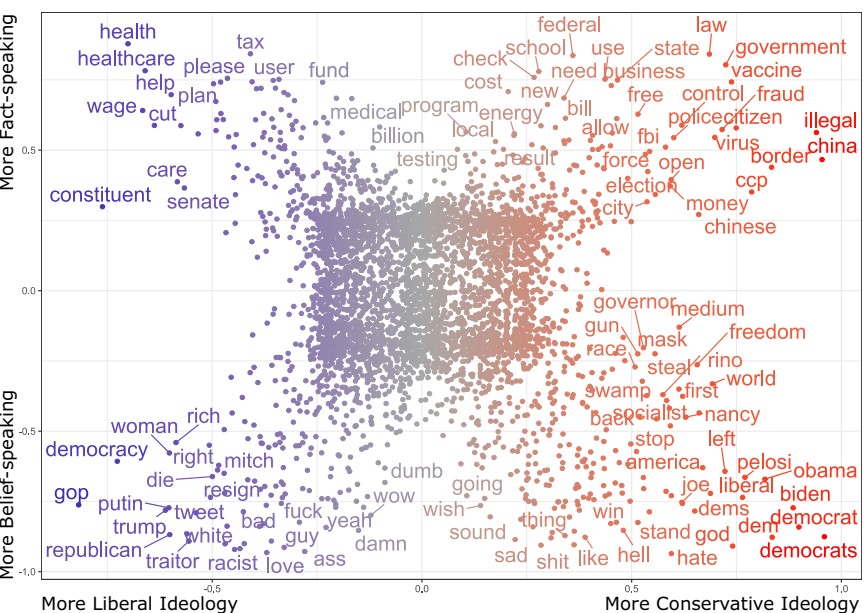

**Fig. 4 | Distribution of keywords of the replies in a textual scatterplot.** Every term is a dot with two coordinates associated with the user ideology score ($x$-axis) and keyness of a word in determining the honesty component ($y$-axis). Each axis represents a Scaled $F$-Score (SFS) ranging from −1 to 1 (see "Methods" for further details). The word color signals how representative the term is for the two ends of the ideological spectrum (i.e., red = more conservative; blue = more liberal). We only show word labels where SFS > 0.5 or SFS < −0.5 for readability reasons.

when these tend towards fact-speaking, compared to Republican seeds. The latter interaction, depicted in Fig. 5b, replicates the "cross-party" effect observed in the analysis of the alignment of honesty constructs. Polarizing language towards Democrats' [Republicans'] seeds is higher when repliers are more [less] conservative. Finally, Date

also has a significant although small positive correlation on Pol$_r$ ($t$(97, 510) = 2.760, $p$ = 0.005, $\beta$ = 0.002, 95%CI = [0.001, 0.004]), indicating that polarizing language in replies has increased over time. Further statistics and details for the model and its variables are presented in Table 1 and in "Methods".

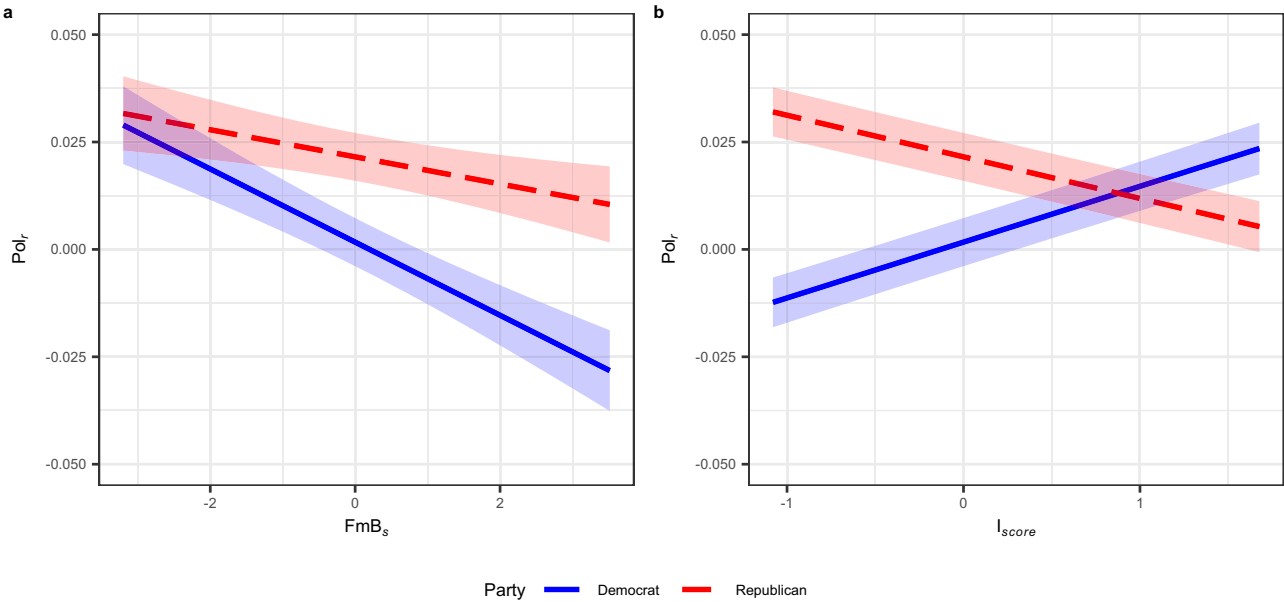

**Fig. 5 | Interactions of seed honesty with party affiliation and ideological alignment on affective polarization.** Significant interactions between the honesty scores of the seeds (FmB$_s$) and the party of the politicians who shared them (Party) in Panel **a**, and between the standardized ideology scores of the repliers ($I_{score}$) and Party in Panel **b**. The *y*-axis shows the affectively polarized language of the replies (Pol$_r$) in both panels. Panel **a** shows how Pol$_r$ decreases more in fact-speaking seeds by Democrats when compared to Republican seeds. Panel **b** indicates that Pol$_r$ towards specific party seeds decreases when the repliers' ideology aligns with the party's, and increases when it does not. The estimates are derived from the regression described in Equation (3) (see "Methods"). The line-types and colors are mapped on the party of the politicians who composed the seeds. The red dotted line represents seeds by Republicans, whereas the blue solid line indicates seeds by Democrats. The shaded areas around the lines represent the 95% confidence intervals.

## Experimental validation of the observational analysis

Our analysis of the Twitter corpus revealed a "contagion", where the honesty conception expressed by politicians in their initial seeds aligned with the honesty conception present in subsequent replies. This contagion occurred even when the same pairs of individuals were involved in a conversation on different occasions. Nonetheless, those results are inescapably correlational because we exercised no control over stimuli or participants. Therefore, to check the robustness of our findings, we decided to test the contagion, and the potential causal role of the seed in determining the tenor of the conversation, in an experimental setting. We preregistered a study in which we invited participants (*N* = 394) to freely reply to synthetic political tweets, as they would on social media. The tweets used as seeds were generated by AI (specifically, Claude AI Sonnet) and covered ten major political issues in the US. The AI generated both a belief-speaking and a fact-speaking version of each seed. Further details about the prompts used and the seeds generated are presented in the Supplementary Information (Section S3). Each participant saw only one version of a seed for each topic, allowing us to control for the honesty conception in the seed while keeping all other variables (e.g., topic, emotions) constant. We then employed a linear mixed-effects model with the FmB scores of the participants' replies as the dependent variable. We specified the FmB scores of the seeds, the E2IS score[42] (a measure of epistemic preference, see "Methods"), and the effectively polarized language of the seed (Pol$_s$) as predictors. Further details about the experiment and its analysis are described in Methods.

The results, reported in Table 2, show that the contagion persisted in the experiment ($t(3, 918) = 5.046$, $p < 0.001$, $\beta = 0.182$, 95%CI = [0.111, 0.253]). Participants' replies tended to follow the same honesty framing that was present in the seed. This finding replicates our basic observation from the Twitter corpus analysis, but because we controlled stimuli here and because participants were randomly assigned to one version of each seed, we can now infer that the nature of the seed caused the subsequent alignment in responses.

This contagion remained significant when controlling for the participants' epistemic preferences (i.e., intuition- vs. evidence-based perspective of truth) and the affectively polarized language of the seed.

We also examined within-subject differences in honesty conceptions in the replies, to check if the same individual would adopt different conceptions on different occasions as determined by the seed.

**Table 2 | Results of the generalized linear mixed-effects models described in Equation (4), with replies' FmB scores (FmB$_r$, left) and replies' affective polarization scores (Pol$_r$, right) as dependent variables**

|  | Dependent variable: | |
|---|---|---|
|  | FmB_r | Pol_r |
|  | (1) | (2) |
| FmB_s | 0.182 | 0.007 |
|  | (0.111, 0.253) | (−0.000, 0.014) |
|  | *p* < 0.001 | *p* = 0.053 |
| EIS | 0.004 | −0.002 |
|  | (−0.042, 0.050) | (−0.007, 0.002) |
|  | *p* = 0.861 | *p* = 0.340 |
| Pol_s | 0.007 | 0.016 |
|  | (−0.071, 0.085) | (0.008, 0.024) |
|  | *p* = 0.858 | *p* < 0.001 |
| Constant | −0.001 | 0.000 |
|  | (−0.107, 0.104) | (−0.015, 0.016) |
|  | *p* = 0.983 | *p* = 0.965 |
| Observations | 3918 | 3918 |

Replies originate from the preregistered experiment described in the Methods section. Estimates are reported together with their 95% confidence intervals. All predictors were scaled by subtracting their means from their values and dividing the result by their standard deviations. The regression was performed with the function lmer from the R library lme4[67].

For this analysis, we averaged each participant's FmB of their replies to belief-speaking and fact-speaking seeds separately. This resulted in two scores for each participant. Then, we conducted two-sided paired $t$-tests by entering each participant's FmB scores for the two different types of seeds into the analysis, examining whether there was a significant difference between the scores. The $t$-tests revealed a significant difference between the two sets of FmB scores ($t(391) = -14.516$, $p < 0.001$, $d = -0.733$, 95%CI of difference in means = $[-0.400, -0.304]$). Participants exhibited higher FmB scores in response to fact-speaking seeds ($M = 0.176$, SD = 0.560) compared to belief-speaking seeds ($M = -0.176$, SD = 0.493).

Our Twitter corpus analysis also revealed that the presence of an evidence-based conception of honesty in the seeds was linked to a decrease in the use of effectively polarized language in the replies. Conversely, the presence of belief-speaking was correlated with an increase in the effectively polarized language in the replies. We tested whether these results also occurred in the experiment. We measured the affectively polarized language of participants' replies (again defined as $Pol_r$). Next, we employed the same linear mixed-effects model described in Equation (4), with the only difference being the dependent variable (i.e., $Pol_r$ instead of $FmB_r$).

The results reported in Table 2, indicate that the effects FmB of the AI-generated seeds on the level of affectively polarized language in the participants' replies were not statistically significant. This finding diverges from our previous corpus analysis of replies to politicians' seeds on Twitter, where we observed a significant and negative association. In contrast, in the experiment, the variable $Pol_s$ had a significant and positive effect ($t(3, 918) = 4.074$, $p < 0.001$, $\beta = 0.016$, 95% CI = $[0.008, 0.024]$), indicating that seeds containing polarizing terms tend to attract more polarized discourse in the replies they receive.

## Discussion

The purpose of this study was to understand how users on Twitter react to two different conceptions of honesty communicated by politicians, one that is based on evidence (fact-speaking) and another that is related to intuition, sincerity, and feelings (belief-speaking). We conducted text analysis of the seeds posted by politicians to see if and how the components in the seeds correlated with the components and the polarization of their replies.

The findings indicate that the components in the replies align with those present in the seeds, regardless of the political parties of the politician and the repliers. This suggests that belief-speaking or fact-speaking seeds tend to elicit similar narrative frames in the replies. However, this "contagion" between politicians and their followers appears to be stronger when there is a same-party affiliation between them. Our findings also show that users who responded to both belief-speaking and fact-speaking messages from the same politicians exhibited alignment in both cases, showing that they adapted their reply to the honesty component in the seed rather than some stable attribute of the seed author. We also found the contagion to persist even when controlling for possible linguistic confounds, such as effectively polarized language, as well as positive and negative emotions (see Supplementary Information, Section S1). Importantly, these observational results were replicated in a preregistered experimental setting. Participants' responses to synthetic political seeds again aligned with the honest framing of the messages they engaged with. This effect persisted even when controlling for participants' epistemic preferences (i.e., their inclination towards intuition or evidence-based perspectives on truth) and the affectively polarized language of the initial seed.

In addition to examining the honesty contagion, we analyzed polarizing language in the replies. We expected fact-speaking to be negatively correlated with affective polarization because of its evidence-based nature. By contrast, belief-speaking is better suited for expressing ideologies and attitudes, as it focuses on emotional content

and streamlines morally charged arguments. Results of the observational analysis highlight the positive association between belief-speaking seeds and the polarized language in the replies compared to fact-speaking seeds. This aligns with findings suggesting that moral-emotional content drives affective polarization[41,43]. The relationship is observable in both parties, and it is particularly noticeable in replies by more conservative users or in replies to seeds written by Republican politicians. By contrast, fact-speaking is negatively correlated with effectively polarized language, especially in the case of replies to Democrats' seeds. However, this result was relatively small, and more importantly, it did not replicate in the controlled experimental settings.

One reason for this could be the extent to which people's attitudes and beliefs are critical factors in how affective polarization is manifested. The key mechanism driving affective polarization is partisan identity, and attitudes and beliefs about political issues serve as signals of such partisan membership[44]. Moreover, attitudes and beliefs contribute to the understanding of affective polarization beyond the content itself, as the structure of these belief systems is predictive of affective polarization[45]. It is possible that an experimental setting does not provide a sufficient condition for participants to engage in a meaningful expression of their partisan identity, thus leaving the affective polarization of the seed they reply to, rather than its belief-fact positioning, as the only predictor for affective polarization.

Overall, these findings support the notion that the tone of online conversations is influenced by the initiating politician. While this "contagion" could apply to other types of framing, such as emotional tweets triggering more emotional responses, the results emphasize the significance of leadership and the role that political elites have in shaping public opinion. Previous research illustrates how the strength and the repetition of frames by politicians have a noticeable influence on how recipients process information[46]. Moreover, the influence of the elite partisan focus over a particular issue, as in the case of climate change, plays a pivotal role in shaping public opinion: increased attention to an issue by political elites triggers amplified media coverage, which, in turn, heightens public concern about that issue[47]. Our conclusions align with and reinforce previous research by showing how politicians who communicate from an evidence-based standpoint can contribute to improving the overall quality of online communication by making it more fact-based.

It is worth noting that, in our previous research, belief-speaking, and fact-speaking were shown to be correlated with news quality and reliability (cf.[15]). More precisely, we found that Republicans are more likely to share lower-quality information compared to Democrats[48], and that this tendency is linked to belief-speaking. The more Republicans engaged in belief-speaking, the more likely they were to share low-quality information. Conversely, an increase in fact-speaking in the tweet corresponded to an increase in information trustworthiness. In contrast, we found little to no statistical evidence of this relationship among Democrats (see also Supplementary Note 7 in ref. 15).

These conclusions do not imply that belief speaking should be avoided by politicians in all circumstances. Political discourse is based on beliefs and ideologies and, at times, such framing may be a relevant and effective means of communication. Neither can we ascertain whether fact-speaking statements by politicians and repliers are, in fact, generally more accurate. Nonetheless, the results from our study suggest that a belief-speaking framing by US politicians on Twitter could lead to a more polarized language compared to an evidence-based one and that employing a fact-speaking narrative may be an effective approach for reducing controversy and mitigating polarized comments, thereby contributing to improved online communication quality.

A limitation of our Twitter corpus analysis is that the observed correlations could not establish a direct causal relationship. However,

our investigation of users who responded to both belief-speaking and fact-speaking seeds from the same politicians suggested a potential causal model. We tested this causal model through a preregistered experiment in which participants were assigned to respond to either fact- or belief-based statements that were identical in all other aspects (e.g., incivility, tone, length, topic). This experimental design allowed us to isolate the impact of fact- versus belief-speaking framings, allowing us to conclude that the nature of the seed caused the subsequent alignment in responses. The causal hypothesis is further buttressed by the finding that the same participants changed their linguistic expressions in response to the seeds, as revealed by our within-participants analysis.

A further limitation of our Twitter corpus analysis pertains to the number of seeds ($N = 10,164$) that remained after the filtering process. Even though this is a small sample, we maintain confidence in the robustness of our conclusions given the extensive number of replies ($N = 97,510$) and, notably, the broad representation of politicians from both parties (Democrats = 386, Republicans = 342) within our sample, which constitutes almost 70% of the accounts in our larger dataset of more than 4 million tweets used in[15]. The fact that we observed the same truth contagion in an experiment with synthetic highly-controlled stimuli further allays concerns associated with the small size of our Twitter corpus.

Finally, we acknowledge the need for caution in presenting claims about the potentially negative association between fact-speaking and effectively polarized language. While the observational analyses initially suggested a relationship, the experimental findings did not replicate this effect, raising questions about its robustness and validity. We also recognize that this failure does not constitute the final word on the matter, and we highlight the need for future research, with improved experimental designs and greater statistical power, to explore whether such an effect could be uncovered.

Future studies should also investigate whether the contagion between leader and user propagates beyond a single conversation, by spreading to other interlocutors, as it happens with emotions[49] and toxicity[50]. Our present study cannot answer that question. Additionally, further research should examine the generalizability of our observed effects on countries with different political systems. Notably, some European regions demonstrate higher affective polarization levels despite having a multiparty system, prompting a deeper investigation[51]. Furthermore, our findings point to practical implications, stressing the significance of accountability in politics. Research highlights that reminding legislators of reputational risks tied to questionable statements can effectively mitigate their negative fact-checking ratings, reiterating the significance of fact-speaking-based framing in political discourse[52].

## Methods

### Data collection

To create our dataset, we chose a random sample of 20,000 tweets published between January 1, 2016 and March 16, 2022, by members of the US Congress as "seeds" for the ensuing conversations with the public. Although the interval of time chosen included Congresses from the 114th to the 117th, for the 114th and 115th Congress, only handles of senators were available. The sample was extracted from a larger corpus of tweets used in one of our previous papers[15]. Data collection for the larger corpus was approved by the Institute of Interactive Systems and Data Science at the Graz University of Technology. In addition, to reduce the chance that the analysis was driven by accounts that post a large number of tweets, we decided to include only the latest 3200 seeds from every account (the default maximum of the Twitter API). For a variety of reasons ranging from deleted seeds/replies, suspended accounts, and parsing errors, we obtained replies from only 13,169 seeds, yielding a total of 331,373 replies. We opted to keep only first-level replies, that is we excluded those that did not address the

initial politician's seed but rather other replies in the conversation. Furthermore, we removed all replies from which we were unable to derive repliers' ideology scores (further details in "Methods"), either because the accounts had been deactivated or because their followers' network was too small. We also removed replies shorter than 10 words. Finally, we opted to keep only one reply per replier, namely the first in chronological order. As a result, the final dataset included 97,510 replies to 10,164 different seeds published by 728 US politicians (Democrats = 386, Republicans = 342).

To measure honesty components in the dataset, we first developed two dictionaries, each comprising of keywords related to the two conceptions of honesty. As an example, keywords for fact-speaking included terms such as "reality", "assess" "examine", "evidence", "fact", "truth", "proof", and so on. For belief-speaking, initial keywords were terms such as "believe", "opinion", "consider", "feel", "intuition", or "common sense". These two lists were expanded using both word embeddings and colexification networks. More specifically, we used the fasttext library[53] to identify and include terms having a cosine similarity greater than 0.75 with the keywords we picked. We additionally expanded the lexicon with the LEXpander method[54] which is based on colexification networks that connect words in a language based on their shared translations to other languages, signaling terms that can communicate similar meanings[55].

Next, we validated the two dictionaries through human annotations in order to observe whether the keywords we chose were pertinent enough for the identification of the two components. This procedure resulted in two distinct dictionaries illustrated in ref. 15.

We then used GloVe[39] to extract word embeddings for each text. Robustness tests performed using different embeddings were performed in ref. 15, displaying similar results. We calculated embeddings for individual terms within each text and then averaged them to create a single 300-dimensional vector that represented each text. This same process was applied to both of our dictionaries, resulting in two separate embedded centroids: one for belief-speaking and another for fact-speaking.

Following ref. 40, we applied the distributed dictionary representation (DDR) approach. We calculated the cosine similarities between the texts' embedded representations and the two dictionaries' centroids. The similarity scores range from −1 (not similar at all) to 1 (perfectly similar). As a result, each text had two scores, one representing its belief-speaking value ($D'_b$) and the other its fact-speaking value ($D'_f$). To account for the influence of tweet length on these scores, we made predictions for each tweet's scores based on its length, and then we subtracted these predictions from both the belief-speaking and fact-speaking similarity scores. To validate the belief-speaking and fact-speaking measures, raters on the Prolific survey platform were asked to score tweets on scales reflecting their representativeness for belief-speaking and fact-speaking, respectively, and the ratings were used to create a ground-truth dataset to compare against the similarity-based classifier, which showed high performance with AUC scores of 0.824 for belief-speaking and 0.772 for fact-speaking (see ref. 15 for further details).

Next, we scaled these two values by subtracting their means from the scores and dividing the result by their standard deviations, and calculated a Fact-minus-Belief score (FmB) for both replies ($FmB_r$) and seeds ($FmB_s$) using the following formula: $FmB = scaled(D'_f) - scaled(D'_b)$. Values of $FmB > 0$ imply that a text engaged predominantly in fact-speaking whereas values of $FmB < 0$ indicate that a text was engaging in belief-speaking.

### Scatterplot keywords

To understand the content of the corpus in terms of honesty conceptions and repliers' ideology, we extracted keywords following the Scattertext approach[56], a Python package designed to illustrate the representativeness (or *keyness*) of terms between corpora.

Starting from raw frequencies, we calculated for each word both the relative frequency across categories (e.g., between Democrats' and Republicans' texts) as well as the relative frequency within a category (e.g., within Democrats' texts). These values are defined by the package author as precision and recall, respectively. The former represents the discriminative power of a word regardless of its frequency in a certain category. For example, a term $t$ might be present only in one of the two parties' texts, therefore being highly characterizing of the party $p$ where the term is present. However, this does not give any indication of its frequency within that party (e.g., it may only appear a few times). That is why we also use a "recall" measure that indicates the percentage frequency with which a word appears in a certain category. We then transformed these two values using a normal cumulative distribution function (CDF) to scale and standardize the scores. Next, we calculated the harmonic mean of the normal CDF-transformed scores, obtaining a Scaled F-Score (SFS), which ultimately is the metric used to identify distinguishing words.

Since keyness is calculated between two different corpora, and in our case both the honesty and ideology scores were continuous variables, we established arbitrary cut-off values to categorize replies along the two categories in a binary fashion. Therefore, repliers with an ideology score $> 0.5$ were labeled as belonging to the "Conservative" corpus, and those with a score $< -0.5$ were labeled as belonging to the "Liberal" corpus. Next, we calculated the SFS of words for both categories, obtaining two values, $SFS^c$ and $SFS^l$. Lastly, we extracted a final SFS that ranges from −1 (more conservative) to 1 (more liberal) using the following formula:

$$SFS = 2 \cdot \left( -0.5 + \begin{cases} SFS^x & \text{if } SFS^x > SFS^y, \\ 1 - SFS^y & \text{if } SFS^x < SFS^y, \\ 0 & \text{otherwise}. \end{cases} \right). \qquad (1)$$

For each term, this formula compares the SFS of the corpus of interest (in our case $SFS^c$) with the SFS of the reference corpus ($SFS^l$). If the former is higher than the latter, then the final SFS score for that term is equal to $SFS^c$. By contrast, if the term has a higher SFS in the reference corpus, then this latter value is kept as a negative score (i.e., $1 - SFS^l$).

We also needed to categorize replies along the honesty dimension, which, in our case, represents a continuum. Therefore, we used quartiles so that replies falling in the 1st or the 4th quartile of the FmB score were labeled as belonging to the "belief-speaking" or "fact-speaking" corpus respectively. Subsequently. we calculated the SFS of words for both categories, obtaining two values, $SFS^b$ (for belief-speaking) and $SFS^f$ (for fact-speaking). Finally, we applied the same formula used for repliers' ideology to extract one single SFS value from $SFS^b$ and $SFS^f$.

At the end of this process, each term had two SFS: one for its ideology distribution and one for its honesty distribution. These two values were used as coordinates for the scatterplot shown in Fig. 4. The scatterplot has an X-shaped structure due to the relatively high number of keywords and the high rate of divergence between the two categories (ideology vs. components). Therefore, the figure presents a dense central cluster consisting of keywords in common across ideology or components, whereas all "outlier" terms are scattered in its four corners.

## Topic modeling

We performed topic modeling on the seeds using the Python package BERTopic[57]. BERTopic utilizes the Sentence-BERT framework to produce embeddings for each document (i.e., tweet) and subsequently decreases the dimensionality of these embeddings using the Uniform Manifold Approximation and Projection (UMAP) technique[58]. Clusters are then identified through HDBSCAN[59], and topic representations are generated using class-based term-frequency inverse-document-frequency (TF-IDF). We chose BERTopic over more established methods like Latent Dirichlet Allocation (LDA) because it is better suited for modeling short and unstructured texts, such as those found in Twitter data[60,61]. Due to BERTopic's reliance on an embedding approach, we minimally preprocessed the data to maintain the original sentence structure, only lemmatizing the entire dataset to produce cleaner topic representations and removing URLs from the texts.

Our approach involved incorporating specific thresholds in the topic modeling process. We established a minimum document frequency of 50 to reduce the number of small topics, opted for 100 neighboring sample points for the manifold approximation to achieve a comprehensive embedding structure representation, and set a minimum document frequency of 5 for the c-TF-IDF to control the topic-term matrix's size and avoid memory-related computational issues. Although one of BERTopic's strengths lies in its ability to determine the number of topics $k$ without prior specification, we aimed to gain insight into the optimal number of topics for our dataset beforehand. To do this, we used ldatuning[62], an R package that uses Latent Dirichlet Allocation to train multiple models and compute their validation metrics. The data that ldatuning modeled was preprocessed by removing stopwords and irrelevant text (punctuation, numbers, URLs, Twitter handles). The results indicated a value of $k$ between 40 and 50 as an optimal number of topics for the dataset. Consequently, we manually set the number of topics to be identified by BERTopic to 40 based on this guidance.

## Honesty alignment analysis

We fitted the following linear regression in order to observe the alignment of honesty constructs across seeds and their replies within a conversation in our Twitter corpus:

$$FmB_r \sim FmB_s \times I_{score} \times Party + Pol_s + Date \\ + Pos + (1|AuthorID/SeedID) + (1|Topic) \qquad (2)$$

Here, $FmB_r$ is the FmB score for the replies, and $FmB_s$ is the FmB score for the politician's seed that initiated the conversation.

We also entered $FmB_s$ in a fully crossed interaction with two further predictor variables, namely the party of the politicians who wrote the seeds (Party), coded as a binary factor ("Democrat" or "Republican"), and the ideology scores of the replies' authors ($I_{score}$). The score is calculated by observing, for each replying account, the political figures they follow on Twitter[63]. To be more specific, each politician whom a replier follows is assigned a partisanship number of −1 or 1, indicating whether the politician is a Democrat or Republican. The average of all partisanship values is then calculated for each replier. The ultimate ideology score ranges from −1, indicating less conservative partisanship, to +1, indicating more conservative beliefs. Figure 6 shows the distributions of our sample of replying accounts across the $I_{score}$ score. Its U-shaped curve suggests that the majority of the accounts only follow political figures belonging to one side of the political spectrum.

The model also included three further predictors: a continuous numeric variable representing the amount of effectively polarized language present in the seeds ($Pol_s$) as we expected belief-speaking seeds to contain more animosity and thus wanted to control for that, a continuous numeric variable representing the position (Pos) of the reply in the conversation in a chronological order, as we expected a conversation to drift further from the original seed as more replies accumulated, and the dates when the seeds were created (Date), reported as objects of class "Date" in R with a "Year-Month-Day Hour:Minutes:Seconds" format, as we wanted to control for a possible temporal effect. Finally, we included three random effects, namely the seeds (SeedID) nested within their authors (AuthorID), as well as the

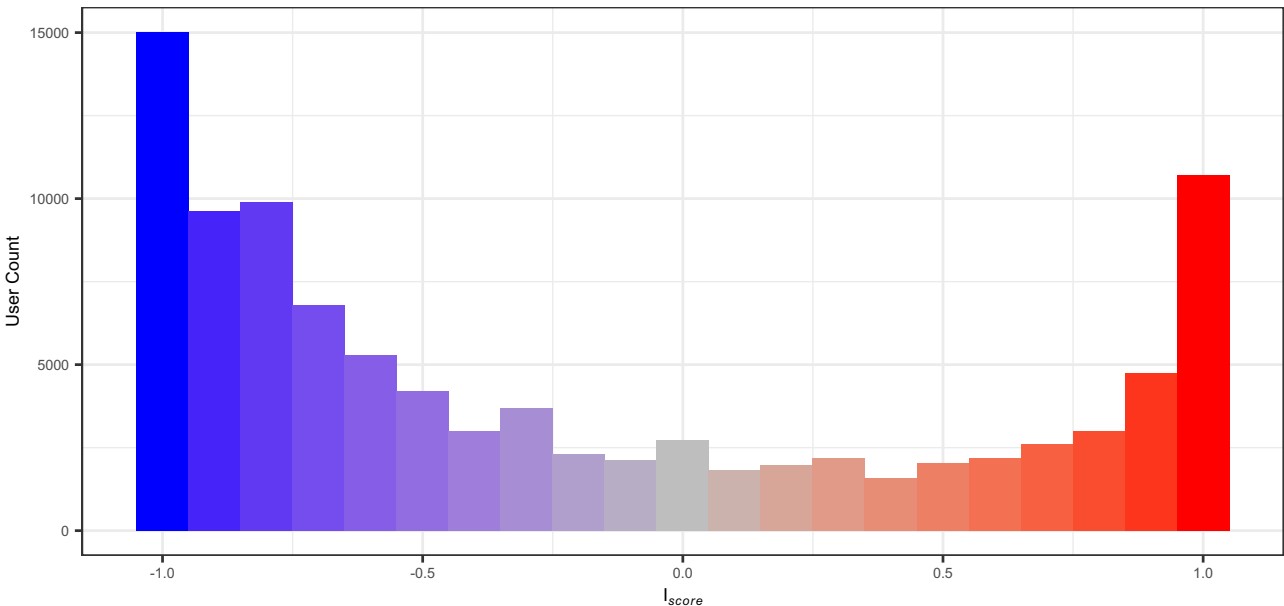

**Fig. 6 | Distribution of users who replied to political tweets (or seeds) in our dataset based on their ideology scores.** The ideology of repliers was calculated by observing the network of politicians they followed on Twitter (now X). $I_{score} = -1$ indicates more liberal accounts (in blue), whereas $I_{score} = 1$ indicates more conservative accounts (in red).

topics (Topic) of the seeds, which we previously classified through topic modeling. All independent variables were standardized (i.e., we subtracted their means and divided them by their standard deviations) before regressing to facilitate interpretation of the results. Data was assumed to follow a normal distribution for the analysis. Post-analysis evaluation of model assumptions through diagnostic plots supported this decision: $Q$–$Q$ plots showed adequate normality of residuals. Given our large sample size, normality tests such as the Kolmogorov-Smirnov and Shapiro-Wilk tests were avoided, as their sensitivity to sample size often flags minor deviations from normality as significant, even when such deviations are unlikely to impact model validity[64,65].

**Polarization analysis**

We measured the presence of polarizing language in the replies by applying the DDR approach[40] used to identify the honesty conceptions to an online polarization dictionary[41]. The dictionary was created by examining the vocabulary used in online communications that show partisan bias. The dictionary also subdivides its keywords based on whether they signal either issue polarization or affective polarization. For our purposes, we only considered those related to affective polarization. We extracted word embeddings for each of those keywords and created an embedded centroid by averaging the embeddings. Then, we calculated the cosine similarity between each of the texts (i.e., seeds and replies) and the centroid representation of the affective polarization dictionary. The similarity scores, renamed $Pol_r$ for replies' polarization scores, range from -1 (not similar at all) to 1 (perfectly similar). We extracted the same measure for the seeds as well, naming them $Pol_s$, as we expect effectively polarized seeds to attract more polarizing language in the replies they receive. As an example, the seed with the lowest polarization score in our dataset ($Pol_s = -0.64$) is the following:

RELEASE: Wagner Introduces Bill to Expand Access to Telehealth Services Read more here

On the other hand, the seed with the highest polarization score in our dataset ($Pol_s = 0.365$) is the following:

When politicians push soft on crime policies and treat police like criminals, violence follows.

Next, we fitted the following regression model:

$$Pol_r \sim FmB_s \times I_{score} \times Party + Pol_s + Date + Pos + (1|AuthorID/SeedID) + (1|Topic) \tag{3}$$

Mirroring the honesty alignment regression, we entered the three-way interactions between the original seeds' FmB scores ($FmB_s$), the politicians' parties (Party), and the repliers' ideology scores ($I_{score}$) to check the association with the replies' polarization scores ($Pol_r$). The variables in the interactions were fully crossed, meaning that, in addition to their three-way interaction, the two-way interactions between them were also included. We also controlled for the polarization scores of the seeds ($Pol_s$), as well as the dates (Date) when the seeds were created. Finally, we also included both the seeds' and their authors' IDs, as well as the topics they related to, as random effects. All independent variables were standardized (i.e., we subtracted their means and divided by their standard deviations) before regressing to facilitate results interpretation. Data was assumed to follow a normal distribution for the analysis. Post-analysis evaluation of model assumptions was conducted through diagnostic plots. While $Q$–$Q$ plots revealed a minor deviation, with a slight hump along the diagonal reference line, the residuals overall adhered to a normal distribution. Similar to the honesty alignment analysis, normality tests, such as the Kolmogorov-Smirnov or the Shapiro-Wilk tests, were not employed due to their sensitivity to large sample sizes, which can result in the over-detection of minor deviations[64,65].

**Truth contagion experiment**

We tested the conversational alignment results from the Twitter corpus analysis in a preregistered experimental setting. Participants ($N$ = 394) were sampled from the United States ($M_{age}$ = 41.98, $SD_{age}$ = 14.16). At the start of the experiment, they were asked to self-report their gender. Our final sample included 175 males, 212 females, 5 individuals who identified as non-binary or other, and 2 individuals

who preferred not to disclose their gender. No statistical method was used to predetermine sample size, as we did not have pre-existing studies that could inform a power analysis to determine a sufficient number of participants. All participants were recruited from the Prolific panel and provided informed consent via mouse click prior to their participation. The study lasted approximately 20 minutes, and the participants received £3 as compensation. The experiment was reviewed and approved by the School of Psychological Science Research Ethics Committee at the University of Bristol (ethics approval #19128).

During the experiment, participants were asked to write a reply to the seeds they were presented with. To ensure that the seeds differed only in honesty framing while keeping other covariates constant, we used Generative AI to create them. We chose Claude AI (version Sonnet 20240229) because it generated higher-quality seeds compared to other Generative AIs. The AI was asked to choose 10 relevant political topics in the US and generate four tweets for each topic, differing only in honesty framing (belief-speaking or fact-speaking) and political stance (in favor or against the topic). Further details about the prompts used and the seeds generated are presented in the Supplementary Information (Section S3).

We ran the prompts starting with a temperature of 0, which is a parameter that controls the randomness of a model's predictions during text generation and increased it by 0.25 increments up to a temperature value of 1. We then selected the most suitable seeds from these runs by measuring their FmB scores and choosing, within each topic, the seeds with the highest and lowest FmB scores. This returned seeds characterized by the prominence of either fact-speaking or belief-speaking. In total, we had 4 seeds per each of the 10 topics, stratified by honesty components (belief-speaking or fact-speaking) and author stance (favoring or opposing the topic).

Seeds were described as "fictional" in the instructions, and their AI-generated nature was revealed in the debriefing. We set a minimum length of 80 characters for each participant's reply to ensure sufficient text for analysis. Each topic randomly displayed one of the four possible seeds. This created a within-participant design, as participants were exposed to both treatments (belief-speaking and fact-speaking seeds). After participants were presented with the seeds, they completed the Evidence-Intuition scale (E2IS)[42], which measures participants' epistemic preferences and provides insight into whether each participant inherently leans towards an evidence-based or intuition-based perspective on truth.

Next, we employed a linear mixed-effects model with the honesty scores of the replies ($FmB_r$) as the dependent variable. The primary independent variable was the honesty score of the seeds ($FmB_s$) to which participants replied. Our random effects included random intercepts for both the different stimuli (i.e., the seeds) nested within the different topics, as well as the different participants. Additionally, we included the E2IS score of each participant, computed by averaging the responses to the Evidence-Intuition scale, and the affective polarization score ($Pol_s$) of the seeds as covariates. Finally, we standardized all independent variables to facilitate the interpretation of the regression estimates.

The following equation illustrates the final regression model formula:

$$FmB_r \sim FmB_s + E2IS + Pol_s + (1|ParticipantID) + (1|Topic\backslash SeedID) \quad (4)$$

The experiment was preregistered at https://aspredicted.org/Y3J_HHM on May 3rd, 2024. We slightly deviated from our preregistration by including the affective polarization scores of the seeds as predictors and adding the seeds' topics as a random effect. Both deviations were justified by the improved performance of the revised model compared to the preregistered one. In the Supplementary Information (Section S4), we report the results for the preregistered model and

demonstrate that they overlap with those presented here. Given the relatively small sample size, normality was assessed using both $Q$–$Q$ plots and the Shapiro−Wilk test. The $Q$–$Q$ plots indicated a normal distribution of residuals, and the Shapiro-Wilk test further supported this assumption ($p = 0.081$), suggesting no significant deviation from normality.

We also conducted an exploratory analysis to examine the relationship between the honesty conceptions in the seeds and the level of affectively polarized language in the replies. We calculated a measure of affective polarization for each reply ($Pol_r$). We then applied the same linear mixed-effects model described in Equation (4), with the dependent variable now being $Pol_r$ instead of $FmB_r$.

**Reporting summary**

Further information on research design is available in the Nature Portfolio Reporting Summary linked to this article.

## Data availability

The lists of Twitter handles of members of Congress used to build the tweet corpus are available from https://www.socialseer.com (114th and 115th Congress), https://doi.org/10.7910/DVN/MBOJNS (116th Congress), and https://triagecancer.org/congressional-social-media (117th and 118th Congress). All the replies analyzed in our studies were collected from a random sample of tweets from the corpus used in ref. 15. The IDs of the tweets from which the replies were collected are reported on OSF[66]. The IDs of the replies texts are also deposited on OSF[66]. Dictionaries of keywords associated with the different conceptions of honesty are deposited on OSF[66]. Dictionaries of keywords used to measure affective polarization are deposited on OSF[66]. Aggregated values for the honesty components and affective polarization of tweets used to produce all figures in this article are deposited on OSF[66]. The data from the preregistered experiment is deposited on OSF[66]. We face constraints in publishing all the essential materials needed for a comprehensive reproduction of our work, particularly concerning tweets. In our context, the tweets encompass both those posted by politicians (which we call "seeds") and the replies they received. Due to data protection considerations and compliance with Twitter's (now X) API usage agreement, revealing the precise textual content of tweets is not feasible. Instead, we offer datasets containing seed and reply IDs, along with the associated metrics derived from the tweet text used in our study. Users can leverage these IDs to rehydrate seeds and replies, obtaining the original texts, as long as the tweets remain accessible at the time of rehydration and the Twitter API v2 still supports tweet rehydration. The necessity to remove the texts from our datasets means that not every aspect of our study can be fully replicated without rehydrating the tweets (see "Code Availability" for further details).

## Code availability

The OSF repository[66] also comprises the scripts utilized to analyze the data and generate the respective visualizations. Instructions to run the scripts are also provided. Scripts requiring tweet rehydration to function correctly pertain to the reproduction of Fig. 4 in the main manuscript, as well as the analyses presented in the Supplementary Information (Section S2) and the Supplementary Fig. 1.

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

## Acknowledgements

This report was partly funded by the John Templeton Foundation through a grant awarded to Wake Forest University for the "Honesty Project". F.C. and S.L. acknowledge funding from the Volkswagen Foundation (grant "Reclaiming individual autonomy and democratic discourse online: How to rebalance human and algorithmic decision making"). S.L. was also supported by funding from the Humboldt Foundation in Germany, and S.L. and D.G. are beneficiaries of the ERC Advanced Grant PRODEMINFO (101020961). J.L. was supported by the Marie Skłodowska-Curie grant No. 101026507. S.L. also receives support from the European Commission (Horizon 2020 grant 101094752 SoMe4Dem) and from UK Research and Innovation (through EU Horizon replacement funding grant number 10049415). The funders had no role in study design, data collection and analysis, the decision to publish, or the preparation of the manuscript.

## Author contributions

F.C., S.L., A.S., and J.L. conceptualized the research. J.L. and S.T.A. collected and curated the data. S.T.A. and F.C. performed computational measurements and statistical analyses. D.G. and A.S. provided advice on the statistical analyses. F.C. administered the preregistered experiment and analyzed its results. F.C. prepared the visualizations. S.L. and D.G. acquired funding and supervised the project. F.C. and S.L. wrote the original draft of the article. All authors contributed to editing the original draft of the article.

## Competing interests

The authors declare no competing interests.
