## [Transparent Peer Review file · Nature Communications]

Different Honesty Conceptions Align Across US Politicians' Tweets and Public Replies

Corresponding Author: Dr Fabio Carrella

Version 0:

Reviewer comments:

Reviewer #1

(Remarks to the Author)

This paper analyzes public responses to political “fact-speaking” and “belief-speaking” on Twitter. The paper provides a valuable contribution of showing how the increase in belief-speaking among politicians begets more belief-speaking and may increase partisan animosity. However, I also have multiple suggestions on how to further improve the paper.

First, the paper could further clarify the definitions of fact-speaking versus belief-speaking. The descriptions and text dictionaries seem to include two major components of belief-speaking— feelings and opinions. These are related but distinct elements, and it may be worthwhile to control for emotional language in the tweets to parse them apart. Another way that fact-speaking and belief-speaking are distinguished in the introduction is that fact-speakers update their beliefs with new information whereas belief-speakers hold onto opinions regardless of facts. However, this can't really be measured in analyses of individual tweets (politicians using belief-speak in a tweet may change their minds and those using fact-speak may support positions with evidence while ignoring counterevidence). A more concrete definition of these two kinds of speech is needed.

Relatedly, many of the belief-speaking key words among both Democrats and Republicans are individual politicians on the other side (e.g. trump, mitch, joe, pelosi). How often are the belief-speaking examples attacks against other politicians? Additionally, how often is belief-speak about something subjective like disliking someone, versus opinions about concrete events which could be verified? If most belief-speak is subjective opinion, it would make sense that the replies would also use belief-speak.

In the introduction, the authors explain that belief-speaking can be used both intentionally or unintentionally, and then give recent examples of politicians using belief-speaking intentionally. It's very likely that politicians do use belief-speaking strategically without actually believing what they say, but rarely can people actually know what politicians think is true, so it is probably best to not make assumptions about when politicians have intentionally lied vs. unintentionally spread misinformation.

Donald Trump is a well-known example of belief-speaking and misinformation in online political debate. However, naming other politicians who subsequently engaged in more belief-speaking rhetoric could be more convincing in portraying the contagion effect and showing the growing pervasiveness of belief-speak in politics.

The paper could benefit from a deeper explanation into the link between belief-speaking and misinformation risk. The paper cites earlier work by the authors which found that there was an association between belief-speaking and lower trustworthiness of sources, but only for Republicans and not Democrats. The authors then interpret, “The findings are compatible with the idea that a distinct conception of honesty that emphasizes sincerity over accuracy can be used by politicians as a gateway to the sharing of low-quality information.” However, based on these results, that interpretation only appears to be the conclusion among Republican politicians. Given that misinformation is one of the major negative consequences the paper gives for belief-speaking, the paper should either measure misinformation directly or more clearly expand on the link between the two from past work.

It would be interesting to see how much engagement fact-speak vs. belief-speak tweets receive. If belief-speak tweets get more likes and retweets, that would further support contagion as a consequence.

Methodologically, I'm not sure that using a difference score to examine belief and fact speak is the best approach. Does this measurement differentiate tweets low in both forms of speech versus those high in both forms of speech? Examining the impacts of tweets that use both strategies would be informative. Politicians may use fact-speak to make their belief-speak seem more credible.

Finally, the effect sizes should be better contextualized. For example, the ideology of the replier and party are very small effects. Overall, the magnitudes of the effects should be better explained to make a strong case for belief-speak increasing polarization.

(Remarks on code availability)

Reviewer #2

(Remarks to the Author)

In "The 'Truth Contagion' Effect in the US Political Online Debate," the authors leverage Twitter data to argue that the distinction between "belief speaking" and "fact speaking" matters for contemporary discourse, including widespread concerns about misinformation. While the paper marshals an impressive amount of evidence, I think a) it is conceptually misguided and b) does not offer sufficiently rigorous tests of its claims, leaving it unsuitable for publication.

The fundamental conceptual problems begins with the distinction between "belief speaking" and "fact speaking." Political speech often mixes the two; to imagine there were once days when politicians spoke mostly in the latter, but not the former, is to engage in a kind of wishful thinking that is not supported by most available evidence. Political speech often involves a factual claim and a belief claim---global temperature is rising, so we must do X, the crime rate is increasing, so we must do Y, with X and Y following from beliefs. While I am willing to grant that the architecture of Twitter (i.e., the character limit) forces people to make claims that are more fact- or belief- based, it does not follow that political speech in general allows for these concepts to be separable. To their credit, the authors seem aware of the limits of this distinction, as they attempt to preempt concerns, writing: "These conclusions do not imply that belief-speaking should be completely avoided by politicians in all circumstances." This seems like a case of too little, too late; the paper seems motivated to prize one form over the other, when in fact both are part and parcel of political communication. I must note here the dearth of references to political science texts that engage at length with the effects of speech and rhetoric on co-partisans (e.g., Lenz 2012). It's not as if merely adding in such references would placate my concerns; rather, I think a fair reading of them would show that the distinction offered here is not particularly meaningful (and leads, inevitably, to excerpts like the above).

There are two other conceptual problems. First, I was entirely uncertain about how the distinction mattered for misinformation concerns. Of course, one kind of speech is fact-based, the other normative, and this should lead to the latter reducing belief in misinformation (or not increasing); but this seems tautological. Second, I couldn't follow the "social contagion" claim. Again, more engagement--not just mere cites!--with relevant literature on this, especially regarding the enormous effect that political leaders have on co-partisans, would be helpful.

Even if I were persuaded by the conceptual claims advanced in this paper, I would remain unconvinced by the evidence. The reason is simple: The authors are proposing a causal model but do not have evidence of causation. Although they concede this point, that is not enough. Pre-registered experiments that test out the theory, including the social contagion component, are needed. At best, the existing paper reads like the introduction, followed by observational work, of a paper centered around randomized trials. That paper would be more compelling. But it is not the paper before us.

I hope the authors revise the manuscript considerably, paying careful attention to a broader swath of literature and acquiring experimental evidence, before submitting elsewhere.

(Remarks on code availability)

The code is cleanly written; I did not run it on my own machine.

Reviewer #3

(Remarks to the Author)

The authors describe two conceptual models of honesty—"fact speaking" and "belief speaking." This is a rhetorical frame being seen more and more with especially after Trump's presidency. The authors find that the presence of fact-speaking or belief-speaking components in politicians' tweets are reflecting in the replies, indicating a contagion effect where the honesty framing of the original message influenced the language of responses.

The paper offers some important and intriguing downstream implications as well. Belief-speaking texts were found to provoke more polarized language in replies compared to fact-speaking seeds, along with a symmetric response to outgroups.

Overall, the framing of the paper could be made a bit more concise, centered around the interaction between source, replier, ideology, and party.

Things to add or justify:

- Why use GloVe for sentence embeddings when BERTopic is used. Directly after? Or are there any robustness tests done with alternative embeddings?
- Are outgroup responses stronger for fact-based versus belief-based? For Figure 5B), I thought L score ranged from -1 to 1, so why would the regression extend past 1? While the opposite directions make sense, the figure would imply responses to Republican tweets are almost always more polarizing—since they intersect at Ideology = 1, polarization in the replies are actually equal for Democrats and Republicans.
(This aligns with recent studies on asymmetric response to out-group toxicity <https://osf.io/preprints/osf/x59qt>).
- Outline how this diverges from existing paradigms of topic versus incivility would be helpful. Additionally, why wasn't it added as a control as well? Prior studies have found it the most crucial to generating engagement. It might be a useful test to run.
(<https://www.pnas.org/doi/full/10.1073/pnas.2024292118>)
- To use the term contagion would require some more justification while this could propagate beyond dyadic engagement patterns beyond elites and their repliers. Unlike retweeting, there isn't a sharing mechanism (especially in the earlier days of Twitter). As the authors cite, Twitter is known for "broadcasting," and the extent that this contagion travels beyond leader-followers is unknown. Citing something like two-step flow could help justify this.

(Remarks on code availability)

Reviewer #4

(Remarks to the Author)

This paper draws a fascinating distinction between fact-based and belief-based forms of honesty and shows that tweets (posts) on Twitter (X) that are more fact-based tend to receive similarly fact-based responses whereas tweets that are more belief-based tend to receive belief-based responses. The paper suggests that this is evidence of social contagion and shows that there are consequences to this piling on effect: fact-based statements tend to "lower the temperature" of discussion (lowering affective polarization) whereas belief-based statements tend to "crank up the heat".

The thesis is interesting, useful to know, and should appeal to a broad audience concerned about the fragility of democracy and what can be done to preserve it. Other noteworthy strengths include the real-world sample of tweets by actual politicians and responses by their actual followers, the attempt to rule out self-selection as an alternative explanation (to be continued below), and the excellent writing.

Three limitations need addressing.

1. Validity of belief-versus-fact statements. To my knowledge, the manuscript did not cite or demonstrate validation evidence of the key measure (belief-versus-fact based statement). Moreover, analyses did not control for other key linguistic variables such as emotional tone or incivility. The examples in Fig 1 seem to have both tone and incivility confounds. Without these analyses, interpreting the main analysis is mired by confounds.
2. Causal claims from correlational data. The paper claims that it establishes that social contagion. Contagion is a causal process, where the presence of the originator causes a change in the receiver. Establishing causation from correlational data is difficult. I applaud the authors for addressing the possibility of self-selection (that people self-select into following and replying to like-minded politicians). The analysis showing that the same people responded differently to belief and fact-based statements was a good start toward establishing contagion, but insufficient because it rules out self-selection at the person level but not at the mindset level. It remains possible that when in a fact-based mindset, people will respond to fact-based statements on Twitter and vice versa for belief-based mindset. That's the thing about correlational data and confounds—it's an unending game of whac-a-mole. To establish the causal process of contagion, experimental evidence is required where participants are randomly assigned to respond to either fact- or belief-based statements that do not differ in any other meaningful way (e.g., incivility, tone, length, content). Absent said data, the interpretation of the present dataset from Twitter/X is tenuous. This applies both to the contagion results as well as the affective polarization results.
3. Theoretical rationale. A more minor suggestion is to strengthen the theoretical reasoning for predicting that fact-based statements will lower affective polarization and vice versa for belief-based statements.

(Remarks on code availability)

Version 1:

Reviewer comments:

Reviewer #1

(Remarks to the Author)

The authors did a good job of addressing my previous concerns.

(Remarks on code availability)

Reviewer #2

(Remarks to the Author)

I appreciate the revisions made to this article. I especially appreciate the fact that the authors went ahead and tested the truth contagion claim via a robust experiment. However, the fact that they could not replicate the polarizing effect of truth-telling experimentally is revealing and calls for additional changes to the manuscript. Most importantly, they should remove claims about this "effect." The claims occur throughout the manuscript but appear concentrated in Section 2.2. This must be done for the following two reasons. First, as a general matter, if an effect cannot be created experimentally even if it appears in some observational analyses, there is reason to doubt its validity. (This is doubly true when other effects that are gleaned from observational work do replicate.) Second, the failure to generate this effect in the lab *should not* be the last word on the issue. It is possible that, with a better-powered, better-designed experiment, the effect would occur. But because the authors have--with great transparency, to their credit--reported not finding the effect, it behooves them, for the sake of cumulative knowledge, not to present the results as is. Doing so would amount to "accepting the null." My advice would be to remove Section 2.2 and all claims about the polarizing effects of truth-telling. Barring that, they could present a condensed version of the observational analyses and the experimental evidence on this specific question in the Discussion section, describing them both as a springboard for future research. If either of those approaches are taken, I would support publication.

(Remarks on code availability)

Reviewer #4

(Remarks to the Author)

This revision is terrific! The authors thoroughly and convincingly addressed all of my comments. I appreciate the clarification of the validity of the main measure. The addition and statistical control of other linguistic variables strengthens the conclusions in the Twitter analysis. Adding experimental evidence that triangulate with the observational findings was particularly strong. And I applaud the addition of more theoretical reasoning for the main prediction. My earlier positive comments remain. This is an interesting, novel, and important paper and I support its publication.

(Remarks on code availability)

Response to reviewers

We thank the reviewers for their insightful comments that helped us to improve the quality of the paper.

In the following, we respond to reviewers by providing the original reviewer's comments in *italic* text and our respective answers in **bold** font. Text additions to the manuscript are indicated by their corresponding line numbers in the main paper.

Reviewer #1 (Remarks to the Author):

This paper analyzes public responses to political “fact-speaking” and “belief-speaking” on Twitter. The paper provides a valuable contribution of showing how the increase in belief-speaking among politicians begets more belief-speaking and may increase partisan animosity. However, I also have multiple suggestions on how to further improve the paper.

We sincerely thank the reviewer for their detailed comments on our manuscript. Their insights were invaluable in identifying its weaknesses and suggesting improvements to enhance the overall quality of the paper.

First, the paper could further clarify the definitions of fact-speaking versus belief-speaking. The descriptions and text dictionaries seem to include two major components of belief-speaking– feelings and opinions.

(1) *These are related but distinct elements, and it may be worthwhile to control for emotional language in the tweets to parse them apart.*

We have added affectively polarized language to the main regression model (i.e., the model that examines the contagion effect). Additionally, we included textual measures of negative and positive emotions in a further regression reported in the first section of the Supplement. Notably, the contagion effect remained significant when controlling for both affectively polarized language and emotions. Furthermore, we found that affective polarization in the seed had an effect on both the honesty conception and the affective polarization of the reply. In the first case, the correlation was negative, indicating that an increase in affective polarization in the politicians' seeds led to more belief-based responses by the repliers. In the second case, the effect was positive, meaning that the presence of affective polarization in the seeds led to an increase in affective polarization in the replies. Finally, the model that included negative and positive emotions did not reveal any significant effects of these emotions.

(2) *Another way that fact-speaking and belief-speaking are distinguished in the introduction is that fact-speakers update their beliefs with new information whereas belief-speakers hold onto opinions regardless of facts. However, this can't really be measured in analyses of individual tweets (politicians using belief-speak in a tweet may change their minds and those using fact-speak may support positions with evidence while ignoring counterevidence). A more concrete definition of these two kinds of speech is needed.*

We addressed Reviewer #1's comment by updating the definitions of belief-speaking and fact-speaking (Line 75). To clarify, the original distinction of honesty into several components was made by Cooper et al. (2021). In that paper, the authors described a component called "truth-seeking" as the ability of a person to "seek out truthful information and update their beliefs based on this information." We aimed to capture this aspect by including related keywords in our fact-speaking dictionary (e.g., correction, rectify, verify).

(3) Relatedly, many of the belief-speaking key words among both Democrats and Republicans are individual politicians on the other side (e.g. trump, mitch, joe, pelosi). How often are the belief-speaking examples attacks against other politicians? Additionally, how often is belief-speak about something subjective like disliking someone, versus opinions about concrete events which could be verified? If most belief-speak is subjective opinion, it would make sense that the replies would also use belief-speak.

We do believe that attacks against other politicians are mostly conveyed using belief-speaking rather than fact-speaking—and the polarization analysis seems to confirm this assumption—however, it does not necessarily follow that replies must also be framed in the same honesty conception. It is entirely possible, for example, that counterpartisans respond to belief-speaking attacks by drawing attention to evidence that defangs the attack, thus employing fact-speaking. Therefore, the contagion we observed is particularly interesting because it suggests that replies are not constrained to be belief-speaking, but rather counter this expectation.

4) In the introduction, the authors explain that belief-speaking can be used both intentionally or unintentionally, and then give recent examples of politicians using belief-speaking intentionally. It's very likely that politicians do use belief-speaking strategically without actually believing what they say, but rarely can people actually know what politicians think is true, so it is probably best to not make assumptions about when politicians have intentionally lied vs. unintentionally spread misinformation.

In the case of Donald Trump, we updated the paragraph to state that we cannot infer the intentionality of politicians using misinformation (not specifically belief-speaking) to divert media attention (Line 126). However, we also provided further evidence suggesting that Trump might intentionally share misinformation to divert media attention (Line 129). Specifically, other studies have identified significant linguistic distinctions between Trump's factually accurate and inaccurate tweets, implying that Trump's inaccurate tweets are unlikely to be random errors and may have been crafted more deliberately.

(5) Donald Trump is a well-known example of belief-speaking and misinformation in online political debate. However, naming other politicians who subsequently engaged in more belief-speaking rhetoric could be more convincing in portraying the contagion effect and showing the growing pervasiveness of belief-speak in politics.

We agree that this is an intriguing possibility but we find it pragmatically intractable because it is difficult to match other politicians' speech to, say, Trump's. We know of no easy way to identify replies to Trump in a public arena. By contrast, on Twitter/X, there is no ambiguity about when people reply to a politician and we therefore focused our analysis on that medium.

The paper could benefit from a deeper explanation into the link between belief-speaking and

misinformation risk. The paper cites earlier work by the authors which found that there was an association between belief-speaking and lower trustworthiness of sources, but only for Republicans and not Democrats. The authors then interpret, “The findings are compatible with the idea that a distinct conception of honesty that emphasizes sincerity over accuracy can be used by politicians as a gateway to the sharing of low-quality information.” However, based on these results, that interpretation only appears to be the conclusion among Republican politicians.

(6) Given that misinformation is one of the major negative consequences the paper gives for belief-speaking, the paper should either measure misinformation directly or more clearly expand on the link between the two from past work.

We added further evidence to our paper by integrating studies that link belief-speaking and misinformation (Line 107). Specifically, the use of intuition-based epistemology by populist leaders may enhance the cohesion among their supporters, therefore rendering the dissemination of misinformation a distinctive indicator of group affiliation and a preference for instinctual judgments over evidence-based assertions. Consequently, this dynamic fosters the proliferation of additional falsehoods.

(7) It would be interesting to see how much engagement fact-speak vs. belief-speak tweets receive. If belief-speak tweets get more likes and retweets, that would further support contagion as a consequence.

Though this type of analysis goes beyond the scope of our paper, we explored the link between user engagement (i.e., likes, retweets, and replies) and how that relates with the honesty component of the seeds in our sample. We conducted a regression with user engagement as a continuous dependent variable, and the FmB score of the seeds as the main independent variable. We also controlled for affective polarization in the seed, the date when the seed was published, and the party of the politician who authored the seed. Finally, we added random intercepts for the authors, as different politicians have different baselines of followers, and topic of the seeds. We find that there is a significant effect of FmB such that the more a seed expresses belief-speaking [fact-speaking], the more [less] engagement it triggers.

(8) Methodologically, I’m not sure that using a difference score to examine belief and fact speak is the best approach. Does this measurement differentiate tweets low in both forms of speech versus those high in both forms of speech? Examining the impacts of tweets that use both strategies would be informative. Politicians may use fact-speak to make their belief-speak seem more credible.

The measurement does not distinguish between tweets that are low or high in both forms of honesty. However, methodologically, we need to compute a difference score because the two measures of honesty are expected to be correlated—they both relate to the concept of honesty. Therefore, we must create an anchored vectorial representation that spans a continuum derived from the difference between these two conceptions. This continuum allows us to project each individual tweet onto a scale based on their position relative to this difference, giving more nuanced results (for further details, see Grand et al. 2022; <https://doi.org/10.1038/s41562-022-01316-8>).

(9) Finally, the effect sizes should be better contextualized. For example, the ideology of the replier and party are very small effects. Overall, the magnitudes of the effects should be better explained to make a strong case for belief-speak increasing polarization.

We revised the phrasing and the reporting of the models' estimates and rescaled the variable "FmB" (fact minus belief) to better facilitate comparison across predictors.

Reviewer #2 (Remarks to the Author):

In "The 'Truth Contagion' Effect in the US Political Online Debate," the authors leverage Twitter data to argue that the distinction between "belief speaking" and "fact speaking" matters for contemporary discourse, including widespread concerns about misinformation. While the paper marshals an impressive amount of evidence, I think a) it is conceptually misguided and b) does not offer sufficiently rigorous tests of its claims, leaving it unsuitable for publication.

The fundamental conceptual problems begins with the distinction between "belief speaking" and "fact speaking." Political speech often mixes the two;

(1) to imagine there were once days when politicians spoke mostly in the latter, but not the former, is to engage in a kind of wishful thinking that is not supported by most available evidence.

In response to this, we do have supporting evidence, although thus far only available as a preprint, indicating that politicians primarily engaged in fact-speaking. Specifically, our analysis of US Congressional speeches reveals a consistent decline in evidence-based language since the mid-1970s. Importantly, this decline correlates with reduced legislative productivity, heightened partisan polarization in Congress, and increased societal income inequality. More details available here: <https://doi.org/10.48550/arXiv.2405.07323>

(2) Political speech often involves a factual claim and a belief claim---global temperature is rising, so we must do X, the crime rate is increasing, so we must do Y, with X and Y following from beliefs.

We agree that both forms can be present in a tweet. For this reason we adopted a difference score to capture instances where one of the two conceptions of honesty is dominating. We invite Reviewer #2 to read our response to comment n. 8 from Reviewer #1 for further details.

While I am willing to grant that the architecture of Twitter (i.e., the character limit) forces people to make claims that are more fact- or belief- based, it does not follow that political speech in general allows for these concepts to be separable. To their credit, the authors seem aware of the limits of this distinction, as they attempt to preempt concerns, writing: "These conclusions do not imply that belief-speaking should be completely avoided by politicians in all circumstances." This seems like a case of too little, too late;

(3) the paper seems motivated to prize one form over the other, when in fact both are part and parcel of political communication.

Both forms are undoubtedly important facets of political communication. We acknowledge that belief-speaking expressions are a necessary component of political debate, as they convey values and identity. However, if the balance shifts to the point where fact-speaking no longer matters, it becomes a cause for concern. In this context, the preprint we shared in our initial response warns that when belief-speaking becomes too predominant, societies become more polarized and unequal, and Congress becomes less productive.

I must note here the dearth of references to political science texts that engage at length with the effects of speech and rhetoric on co-partisans (e.g., Lenz 2012). It's not as if merely adding in such references would placate my concerns; rather, I think a fair reading of them would show that the distinction offered here is not particularly meaningful (and leads, inevitably, to excerpts like the above).

We are grateful to the reviewer for providing us with this reference. We believe that Lenz's studies on leadership and followership complement our own research rather than rendering it less meaningful. Lenz's work employs different methodologies (such as panel surveys) and focuses on policy support rather than conversational analysis on social media, thus offering a distinct perspective on the topic. Furthermore, while we acknowledge that Lenz's work also addresses the importance of honesty in politicians' performance ratings by followers, our study diverges in its focus on the distinct conceptions of honesty that we have theorized in previous works (Lasser et al., 2023), and how these are measured using our dictionary approach

In this regard, we thoroughly validated our approach in three steps. First, we validated candidate keywords for belief- and fact-speaking by asking participants (N=51) on Prolific to rate the representativeness of candidate keywords for each honesty conception. Keywords significantly more representative of belief-speaking or fact-speaking were included in the respective dictionaries. Second, we applied the dictionaries to a tweet corpus and calculated semantic similarity scores between tweets and the dictionaries. We surveyed Prolific participants again to validate these scores against human ratings, finding satisfactory agreement with AUC=0.824 for belief-speaking and AUC=0.772 for fact-speaking. Third, we applied the dictionaries to New York Times articles in 'opinion', 'politics', and 'science' categories. Science articles were most similar to fact-speaking, followed by politics and opinion articles. Opinion articles showed the highest similarity to belief-speaking, followed by science and politics.

Overall, these validation steps demonstrate that our dictionaries effectively capture and differentiate between the two honesty conceptions—belief-speaking and fact-speaking—in a meaningful way.

There are two other conceptual problems.

(4) First, I was entirely uncertain about how the distinction mattered for misinformation concerns. Of course, one kind of speech is fact-based, the other normative, and this should lead to the latter reducing belief in misinformation (or not increasing); but this seems tautological.

We find it unclear how normative discourse could effectively diminish belief in misinformation, particularly when contrasted with fact-based speech. Nonetheless, we have revised the text to incorporate additional evidence from external studies that illustrate the connection between belief-speaking and misinformation.

(5) Second, I couldn't follow the "social contagion" claim. Again, more engagement--not just mere cites!--with relevant literature on this, especially regarding the enormous effect that political leaders have on co-partisans, would be helpful.

We appreciate the reviewer's emphasis on engaging more deeply with the literature on social contagion, particularly concerning the influence of political leaders on co-partisans. We acknowledge the existence of relevant literature on this matter that describes at length how

leadership greatly influences followers' affective expressions (Cherulnik et al. 2006, <https://doi.org/10.1111/j.1559-1816.2001.tb00167.x>; Erez et al. 2008, <https://psycnet.apa.org/doi/10.1037/0021-9010.93.3.602>), especially in presence of an affective congruency between the two parts (Sy & Choi, 2013, <https://doi.org/10.1016/j.obhdp.2013.06.003>).

Our polarization analysis aligns with these findings, demonstrating that an affectively polarized message tends to elicit similarly affectively polarized language in its responses. Furthermore, our contagion analysis contributes by highlighting that the contagion effect of honesty conceptions can be amplified in the presence of partisanship congruency. Importantly, our analysis also reveals that this contagion effect persists even when such congruency is absent.

(6) Even if I were persuaded by the conceptual claims advanced in this paper, I would remain unconvinced by the evidence. The reason is simple: The authors are proposing a causal model but do not have evidence of causation. Although they concede this point, that is not enough. Pre-registered experiments that test out the theory, including the social contagion component, are needed. At best, the existing paper reads like the introduction, followed by observational work, of a paper centered around randomized trials. That paper would be more compelling. But it is not the paper before us.

We appreciate the author's emphasis on the importance of experiments in testing our contagion theory. In response to the reviewer's critique, we have added a preregistered experiment to our study. In this experiment, we exercised control over several variables such as topic, tone, length, so that participants were exposed to either a belief- or a fact-speaking version of the same tweet. The results of this experiment replicates what we found in our Twitter corpus analysis, and provide stronger causal evidence supporting our contagion theory. We hope that, as a result of these additions, our paper will be perceived as more theoretically robust.

I hope the authors revise the manuscript considerably, paying careful attention to a broader swath of literature and acquiring experimental evidence, before submitting elsewhere.

Reviewer #2 (Remarks on code availability):

The code is cleanly written; I did not run it on my own machine.

Reviewer #3 (Remarks to the Author):

The authors describe two conceptual models of honesty—"fact speaking" and "belief speaking." This is a rhetorical frame being seen more and more with especially after Trump's presidency. The authors find that the presence of fact-speaking or belief-speaking components in politicians' tweets are reflecting in the replies, indicating a contagion effect where the honesty framing of the original message influenced the language of responses.

The paper offers some important and intriguing downstream implications as well. Belief-speaking texts were found to provoke more polarized language in replies compared to fact-speaking seeds, along with a symmetric response to outgroups.

We appreciate the positive stance and thank the reviewer for the insightful comments.

Overall, the framing of the paper could be made a bit more concise, centered around the interaction between source, replier, ideology, and party.

Things to add or justify:

1) Why use GloVe for sentence embeddings when BERTopic is used. Directly after? Or are there any robustness test done with alternative embeddings?

We have updated the Methods section (Line 620) to include information on robustness tests conducted in our previous paper (Lasser et al., 2023), which demonstrated similar results with GloVe embeddings. Additionally, we argue that GloVe represents a preferable choice compared to other embeddings, such as BERT, due to its ability to capture clear semantic relationships between words. Unlike BERT embeddings, which are context-dependent and can vary based on the surrounding text, GloVe embeddings offer a clearer understanding of word relationships, particularly when analyzed in isolation from their context.

2) Are outgroup responses stronger for fact-based versus belief-based? For Figure 5B), I thought I_score ranged from -1 to 1, so why would the regression extend past 1? While the opposite directions make sense, the figure would imply responses to Republican tweets are almost always more polarizing—since they intersect at Ideology = 1, polarization in the replies are actually equal for Democrats and Republicans.

(This aligns with recent studies on asymmetric response to out-group toxicity <https://osf.io/preprints/osf/x59qt>).

We appreciate the reviewer for bringing this to our attention. We have now clarified in the Figure caption that all predictors in the regression models, including the I_score, are standardized. This involves subtracting the average of all ideology scores (M = -0.21) from each score, and then dividing the result by the standard deviation of all ideology scores (SD = 0.72). As a result, the I_score no longer maintains its original range of -1 to 1, but now ranges between -1.08 and 1.67.

Figure 5B illustrates that the “most conservative” repliers exhibit greater affective polarization towards Democrats’ tweets compared to Republicans’ tweets. The intersection at 1, where Democrats and Republicans receive similar affective polarization, corresponds to an original ideology score of 0.5.

3) Outline how this diverges from existing paradigms of topic versus incivility would be helpful. Additionally, why wasn’t it added as a control as well? Prior studies have found it the most crucial to generating engagement. It might be a useful test to run.

(<https://www.pnas.org/doi/full/10.1073/pnas.2024292118>)

We thank the reviewer for the suggestion. Thanks to it, we have followed-up extensively by adding affectively polarized language to the main regression model (i.e., the model that examines the contagion effect). Additionally, we included textual measures of negative and positive emotions in a further regression reported in the first section of the Supplement. Crucially, the contagion effect remained statistically meaningful even after accounting for affectively polarized language and emotions. Moreover, we observed that affective polarization in the initial message influenced both the perception of honesty and the emotional polarization in the responses. Specifically, higher

initial affective polarization correlated negatively with the tendency for replies to be based on factual information. Conversely, it positively correlated with increased emotional polarization in the responses. Lastly, the inclusion of negative and positive emotions in the model did not yield any statistically significant effects.

4) To use the term contagion would require some more justification while this could propagate beyond dyadic engagement patterns beyond elites and their repliers. Unlike retweeting, there isn't a sharing mechanism (especially in the earlier day of Twitter). As the authors cite, Twitter is known for "broadcasting," and the extent that this contagion travels beyond leader-followers is unknown. Citing something like two-step flow could help justify this.

We appreciate the point raised by the author and acknowledge, both here and in the paper's limitations section (Line 566), that the analysis of the extent to which such contagion travels beyond conversational exchanges, similar to toxicity and emotions, falls outside the scope of this paper.

Reviewer #4 (Remarks to the Author):

This paper draws a fascinating distinction between fact-based and belief-based forms of honesty and shows that tweets (posts) on Twitter (X) that are more fact-based tend to receive similarly fact-based responses whereas tweets that are more belief-based tend to receive belief-based responses. The paper suggests that this is evidence of social contagion and shows that there are consequences to this piling on effect: fact-base statements tend to "lower the temperature" of discussion (lowering affective polarization) whereas belief-based statements tend to "crank up the heat".

The thesis is interesting, useful to know, and should appeal to a broad audience concerned about the fragility of democracy and what can be done to preserve it. Other noteworthy strengths include the real-world sample of tweets by actual politicians and responses by their actual followers, the attempt to rule out self-selection as an alternative explanation (to be continued below), and the excellent writing.

We thank the reviewer for their positive and kind remarks, which made their critiques constructive and encouraging.

Three limitations need addressing.

1. Validity of belief-versus-fact statements. To my knowledge, the manuscript did not cite or demonstrate validation evidence of the key measure (belief-versus-fact based statement).

We underlined in the Methods (Line 631) that we performed document-level validation to validate the key measures in our previous paper (Lasser et al. 2023). Specifically, raters on the Prolific survey platform were tasked with scoring tweets on scales representing their representativeness for belief-speaking and fact-speaking. These ratings were utilized to establish a ground-truth dataset for comparison against the similarity-based classifier. The classifier demonstrated strong performance, achieving AUC scores of 0.824 for belief-speaking and 0.772 for fact-speaking.

Moreover, analyses did not control for other key linguistic variables such as emotional tone or incivility. The examples in Fig 1 seem to have both tone and incivility confounds. Without these analyses, interpreting the main analysis is mired by confounds.

We have now added affectively polarized language to the main regression model (i.e., the model that examines the contagion effect). Additionally, we included textual measures of negative and positive emotions in a further regression reported in the first section of the Supplement. The contagion effect remained statistically significant even after considering affectively polarized language and emotions. We also found that affective polarization in the initial message affected both the perception of honesty and the emotional polarization in the responses. Specifically, greater initial affective polarization was negatively associated with the likelihood of replies being based on factual information. Conversely, it was positively associated with increased emotional polarization in the responses. Finally, the inclusion of both negative and positive emotions in the model did not produce any statistically significant effects.

2. Causal claims from correlational data. The paper claims that it establishes that social contagion. Contagion is a causal process, where the presence of the originator causes a change in the receiver. Establishing causation from correlational data is difficult. I applaud the authors for addressing the possibility of self-selection (that people self-select into following and replying to like-minded politicians). The analysis showing that the same people responded differently to belief and fact-based statements was a good start toward establishing contagion, but insufficient because it rules out self-selection at the person level but not at the mindset level. It remains possible that when in a fact-based mindset, people will respond to fact-based statements on Twitter and vice versa for belief-based mindset. That's the thing about correlational data and confounds--it's an unending game of whac-a-mole. To establish the causal process of contagion, experimental evidence is required where participants are randomly assigned to respond to either fact- or belief-based statements that do not differ in any other meaningful way (e.g., incivility, tone, length, content). Absent said data, the interpretation of the present dataset from Twitter/X is tenuous. This applies both to the contagion results as well as the affective polarization results.

We agree with the point made and we thank the reviewer for their suggestion. We integrated a preregistered experiment in our paper to test our hypotheses. More precisely, we recruited 393 participants to freely respond to political tweets, simulating their behavior on social media platforms. These tweets, or "seeds," were generated by AI (specifically, Claude AI Sonnet) and covered ten major political issues in the US. Notably, the AI produced both belief-speaking and fact-speaking versions of each seed. Thus, unlike in the naturalistic study, we exercised control over the content of both versions of the seeds. Each participant was presented with only one version of a seed for each topic, allowing us to control for the honesty conception in the seed while keeping all other variables (e.g., topic, emotions) constant. Subsequently, we employed a linear mixed-effects model with the FmB scores of the participants' replies as the dependent variable. We also controlled for epistemic preferences of participants (intuition- vs evidence-based), as well as for affectively polarized language in the AI generated seeds. We found that the contagion effect remained statistically significant even after controlling for affectively polarized language and emotions. Additionally, we found that the emotional polarization present in the initial message influenced both how honesty was perceived and the emotional polarization observed in the responses.

3. Theoretical rationale. A more minor suggestion is to strengthen the theoretical reasoning for

predicting that fact-based statements will lower affective polarization and vice versa for belief-based statements.

We thank the reviewer for providing us with the opportunity to expand on our theoretical rationale. We have now provided a more detailed explanation of our theoretical reasoning behind predicting that belief-speaking statements will increase affective polarization (and vice-versa for fact-speaking statements). More specifically, we have specified in the discussion that, differently from evidence-based fact-speaking, belief-speaking is better suited for expressing ideologies and attitudes (Line 486). This is because belief-speaking focuses on emotional content and streamlines morally charged arguments. This aligns with findings suggesting that moral-emotional content drives affective polarization (see Brady et al., 2017, <https://doi.org/10.1073/pnas.1618923114>; Simchon et al., 2022, <https://doi.org/10.1093/pnasnexus/pgac019>).

Furthermore, we also integrated studies which show how individuals' attitudes and beliefs are crucial in the manifestation of affective polarization (Line 498). Partisan identity is identified as a primary driver, where attitudes and beliefs regarding political issues act as indicators of partisan affiliation (Dias & Lelkes, 2022, <https://doi.org/10.1111/ajps.12628>). Furthermore, beyond the content itself, the structure of these belief systems predicts affective polarization (Turner-Zwinkels et al., 2023, <https://doi.org/10.1177/01461672231183935>).

Response to reviewers

We appreciate the reviewers for their valuable feedback, which has significantly enhanced the quality of this final draft.

Below, we respond to reviewers by providing the original reviewer's comments in *italic* text and our respective answers in **bold** font.

Reviewer #1 (Remarks to the Author):

The authors did a good job of addressing my previous concerns.

We thank the reviewer for their positive feedback and are pleased to know that we effectively addressed the reviewer's previous concerns.

Reviewer #2 (Remarks to the Author):

*I appreciate the revisions made to this article. I especially appreciate the fact that the authors went ahead and tested the truth contagion claim via a robust experiment. However, the fact that they could not replicate the polarizing effect of truth-telling experimentally is revealing and calls for additional changes to the manuscript. Most importantly, they should remove claims about this "effect." The claims occur throughout the manuscript but appear concentrated in Section 2.2. This must be done for the following two reasons. First, as a general matter, if an effect cannot be created experimentally even if it appears in some observational analyses, there is reason to doubt its validity. (This is doubly true when other effects that are gleaned from observational work do replicate.) Second, the failure to generate this effect in the lab *should not* be the last word on the issue. It is possible that, with a better-powered, better-designed experiment, the effect would occur. But because the authors have--with great transparency, to their credit--reported not finding the effect, it behooves them, for the sake of cumulative knowledge, not to present the results as is. Doing so would amount to "accepting the null." My advice would be to remove Section 2.2 and all claims about the polarizing effects of truth-telling. Barring that, they could present a condensed version of the observational analyses and the experimental evidence on this specific question in the Discussion section, describing them both as a springboard for future research. If either of those approaches are taken, I would support publication.*

We thank the reviewer for their valuable suggestions. We have decided to retain the section on polarization results. Additionally, we have removed all claims of causality and emphasized that the results from the observational analysis are correlational. Finally, we added a paragraph in the limitations section highlighting that the polarization results do not replicate under experimental conditions and used this as a basis for suggesting future research. Specifically, we included the following:

Finally, we acknowledge the need for caution in presenting claims about the potential negative association between fact-speaking and affectively polarized language. While the observational analyses initially suggested a relationship, the experimental findings did not replicate this effect, raising questions about its robustness and validity. We also recognize that this failure does not

constitute the final word on the matter, and we highlight the need for future research, with improved experimental designs and greater statistical power, to explore whether such an effect could be uncovered.

Reviewer #4 (Remarks to the Author):

This revision is terrific! The authors thoroughly and convincingly addressed all of my comments. I appreciate the clarification of the validity of the main measure. The addition and statistical control of other linguistic variables strengthens the conclusions in the Twitter analysis. Adding experimental evidence that triangulate with the observational findings was particularly strong. And I applaud the addition of more theoretical reasoning for the main prediction. My earlier positive comments remain. This is an interesting, novel, and important paper and I support its publication.

We thank the reviewer for the extremely positive feedback.